# Neural computations underpinning the strategic management of influence in advice giving

Uri Hertz [1,2,3,4], Stefano Palminteri [5,6,7], Silvia Brunetti[1], Cecilie Olesen[1], Chris D Frith[8,9] & Bahador Bahrami[1]

Research on social influence has focused mainly on the target of influence (e.g., consumer and voter); thus, the cognitive and neurobiological underpinnings of the source of the influence (e.g., politicians and salesmen) remain unknown. Here, in a three-sided advice-giving game, two advisers competed to influence a client by modulating their own confidence in their advice about which lottery the client should choose. We report that advisers' strategy depends on their level of influence on the client and their merit relative to one another. Moreover, blood-oxygenation-level-dependent (BOLD) signal in the temporo-parietal junction is modulated by adviser's current level of influence on the client, and relative merit prediction error affects activity in medial-prefrontal cortex. Both types of social information modulate ventral striatum response. By demonstrating what happens in our mind and brain when we try to influence others, these results begin to explain the biological mechanisms that shape inter-individual differences in social conduct.

[1] UCL Institute of Cognitive Neuroscience, University College London, 17 Queen Square, London WC1N 3AR UK. [2] School of Advanced Study, University of London, Senate House, Malet Street, London WC1E 7HU, UK. [3] Information Systems Department, University of Haifa, Haifa 3498838, Israel. [4] School of Political Sciences, University of Haifa, Haifa 3498838, Israel. [5] Laboratore de Neurosciences Cognitives, Institut National de la Santé et de la Recherche Médicale, Paris 75005, France. [6] Departement d'Études Cognitives, École Normale Supérieure, Paris 75005, France. [7] Institut d'Études Cognitives, Université de Recherche Paris Sciences et Lettres, Paris 75005, France. [8] Wellcome Trust Centre for Neuroimaging, University College London, 12 Queen Square, London WC1N 3BG, UK. [9] Institute of Philosophy, School of Advanced Study, University of London, Senate House, Malet Street, London WC1E 7HU, UK. Correspondence and requests for materials should be addressed to U.H. (email: uhertz@is.haifa.ac.il)

The role of social influence in our lives cannot be overstated. Critical issues ranging from the outcomes of political campaigns to our stand on issues such as global warming, immigration and taxation depend on people trying to persuade the general public[1]. Social influence is also intertwined with our everyday life, as we try to persuade our children, influence our bosses and gain popularity among our friends. Research on social influence has been dominated by the motivation to understand the minds of the targets of influence—the "clients" (e.g., consumers and voters)—in order to exert even more influence on them[2]. Far less is known about the cognitive and neurobiological processes at play in the persuaders—the "advisers" (e.g., politicians and salesmen). Here we ask what happens in the advisor's brain when engaged in the attempt to influence others.

Bayarri and DeGroot[3] proposed a normative solution for how advisers should modulate their strategy for offering their predictions in order to maximize their influence. They assumed that clients are affected by advisers' accuracy and confidence[4–7], i.e., they will be more likely to follow advisers that express confidence only when it is warranted and discredit highly confident but inaccurate advisers. In this case, in order to be selected by the client, advisers should modulate their advice-giving strategy depending on their current influence on the client. When the adviser is failing to influence the advisor, s/he should express higher confidence than permitted by objective evidence (positive advice confidence deviance). On the other hand, when the adviser is trying to maintain and protect already high influence, s/he should sit on the fence with cautious, nuanced advice that is lower than justified by the evidence (negative advice confidence deviance). We will call this a 'competitive' strategy for acquiring and maintaining influence.

Social rank theory[8,9] proposes an alternative account of advising behaviour by positing that people are not motivated just by the desire to influence others' choices, but also by the fear of being excluded by the target of their influence, their client. This theory suggests that an adviser's confidence should be proportional to his/her rank in a group. Lower rank individuals adopt submissive behaviours, expressing lower confidence and avoiding eye contact[8]. Accordingly, social rank theory suggests that, in contrast to the 'competitive' strategy described above, humans may adopt a 'defensive' strategy to manage influence by giving cautious advice when they are ignored by the client (i.e., when their influence is low) and exaggerating their confidence when their influence is high. In addition, social rank theory underscores the importance of an active process of social comparison by which an adviser evaluates her rank by tracking her performance relative to rival advisers[10–14]. In this view, relative performance or merit may also affect advice confidence leading people to display higher confidence when they think they perform better than their peers.

These theoretical models of strategic advice giving and influence management may rely on a number of social cognitive processes, including mentalizing (theory of mind 'ToM')[15], social motivation[16] and social comparisons[17], which have been previously linked to specific neural substrates[16,18]. To track one's current level of influence (i.e., client preference for each advisor) the adviser needs to evaluate the client's state of mind from his actions, presumably incorporating the brain's ToM[17] circuits. ToM processes, such as mentalizing and updating other's beliefs, have been linked to activity in the right temporo-parietal junction (rTPJ)[19,20] and the medial prefrontal cortex (mPFC)[15,20–23]. Influence management also requires the adviser to track her own and her rival's performance in order to calculate her relative merit and adjust this as new evidence about rival's performance emerges[13]. Previous works suggest that this cognitive process is linked to activity in the mPFC, which tracks rank and social status[24,25], and the performance of collaborators and competitors[26–29]. Finally, the adviser is motivated to increase her influence over the client and her merit over the rival adviser[13]. Brain areas such as the ventral striatum (VS) and the ventro-mPFC[30], which track primary (e.g., food) rewards, are also responsive to secondary social rewards such as being selected by others[31], having an increase in social status[25] and being confirmed by the group[32]. These studies lead us to hypothesize that fluctuations in social status and relative merit are tracked by a reward-sensitive network.

Here we examine how people strategically manage their influence and test the hypotheses we derived above regarding the cognitive processes and neural systems underlying this process. First, we examined the behavioural predictions of a normative model of advice giving[3] against those drawn from social rank theory[8]. We asked whether people would give overconfident ('competitive' strategy[3]) or under-confident ('defensive' strategy[8,9]) advice when they are ignored by their client, and whether social comparison with a rival adviser has a role in advising behaviour. We then sought to identify the neural mechanisms underlying the participants' attempts to influence others by advice giving, examining whether (and how) brain areas associated with mentalizing, social comparison and valuation track the appraisal made by a client and calculate relative performance during a strategic advice-giving task.

We find that advice-giving behaviour is driven by an interaction between the adviser's current level of influence over the client and the accuracy of her advice relative to the rival adviser. In four separate experiments, we show that advisers' advice confidence was highest when recent history of advice and outcomes favoured the adviser over her rival, but the client chose to listen to the rival. Using a model-based functional magnetic resonance imaging (fMRI), we find that trial-by-trial variations in selection by client and relative merit prediction error were tracked in separate cortical regions: the right TPJ and the mPFC, respectively. In addition, we observed that trial-by-trial changes in both variables modulated activity in the VS.

## Results

**Behavioural task**. We devised a social influence scenario in which two advisers competed for influence over a client (Fig. 1, online demo: http://www.urihertz.net/AdviserDemo). On a series of trials, a client is looking for a reward hidden in a black or a white urn. The client relies on two advisers who have access to evidence about the probability of the reward being in the black or the white urn. At the beginning of each trial (appraisal stage), the client chooses the adviser whose advice (given later) will determine which urn the client will open. The client's choice of adviser is displayed to the advisers, who then proceed to the evidence stage. They see a grid of black and white squares for half a second. The ratio between the black and white squares indicates the probability of the reward location. Next step is the advice stage. Each adviser declares her advice about the reward location using a 10-level confidence scale, ranging from 'certainly in the black urn' (score of 5B) to 'certainly in the white urn' (score of 5W). Subsequently, both declarations are shown to both advisers and the client (showdown stage). Finally, at the outcome stage, the urn indicated by the chosen adviser's advice is opened and its content is revealed to everyone. The next trial begins with the client selecting an adviser—the client can decide to switch advisers or to stick with the same adviser.

We were interested in the way advisers use advice confidence as a persuasive signal to manage their influence over the client. In our first experiment, therefore, all participants were assigned to play the role of adviser, whereas the rival adviser and the client's

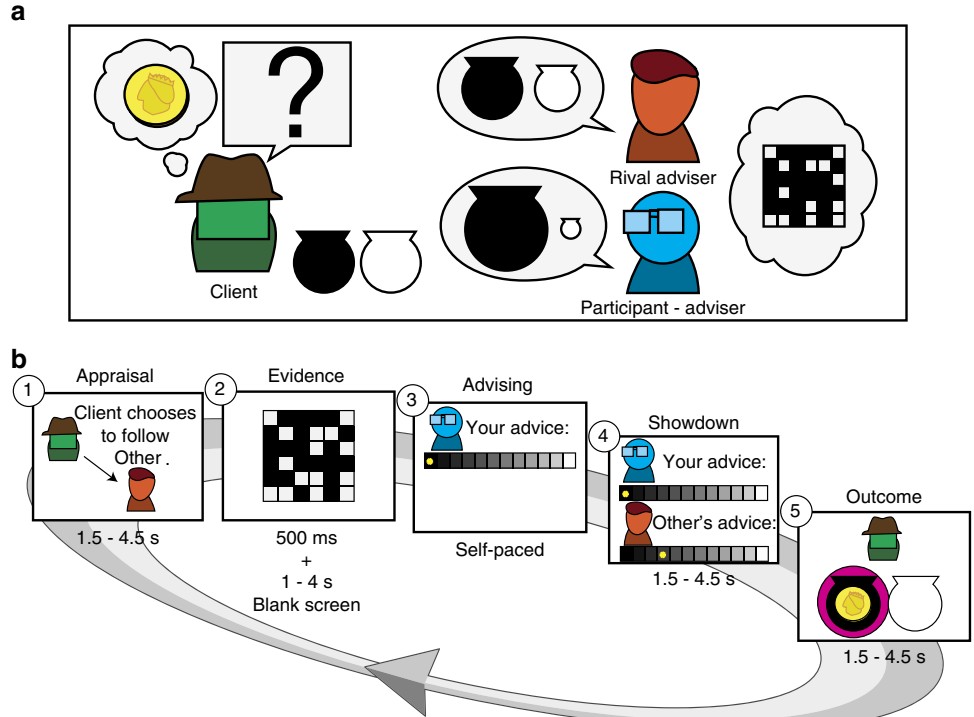

**Fig. 1** Experimental design. **a** Participants were engaged in an advice giving task. In this task, a client is looking for a coin hidden in a black or white urn. He relies on advice from two advisers (the participant (blue) and a computer-generated rival adviser (red)). The advisers, but not the client, have access to information regarding the probability of the coin location. The client considers the advisers' previous success and current confidence when choosing an adviser to follow on each trial. **b** Each trial contains five stages. (1) Appraisal: in the beginning of each trial the client chooses the adviser he is going to follow on the commencing trial (and consequently which adviser is ignored). (2) Evidence: the participant (and the rival) then sees a grid of black and white squares, whose ratio represents the probability of the coin being in the black urn. (3) Advising: the participant states his advice on coin location using a 10 levels confidence scale ranging from 'definitely in the black urn' (5B) to 'definitely in the white urn' (5 W). (4) Showdown: both advisers' opinion is displayed. (5) Outcome: the content of the urn suggested by the selected adviser (magenta circle) is revealed. The next trial starts with appraisal by client based on the history of confidence and success. The stages timings indicated are from the fMRI experiment, where jitter was introduced between stages. See supplementary materials for online and lab stages timings. An interactive demo of the experiment can be viewed at http://www.urihertz.net/AdviserExperiment.html

behaviour were governed by algorithms adapted from Bayarri and DeGroot[3] (see Methods). We examined participants in three independent cohorts: online ($N = 58$), in the lab ($N = 29$) and in-the-scanner ($N = 32$). There were some minor differences between the groups, as online participants performed fewer trials than those in the lab and the scanner. Online participants were paid a bonus for the number of times the client selected them, whereas the cohorts tested in the lab and the scanner received a fixed monetary compensation (see full details in Methods section). Finally, to examine whether the observed behaviour can be generalized to real life social interactions, we ran another lab-based interactive version of the experiment (48 participants organized in 16 triads, thus comprising 32 advisers and 16 clients) in which all roles (advisers and client) were played by participants who genuinely interacted with one another.

**The effect of selection by client and relative merit**. To assess the advising behaviour, we examined the trial-by-trial deviance of advice confidence from probabilistic evidence (Fig. 2a). If the adviser is strictly committed to communicating the information she is given, then confidence exactly matches the ratio of black to white squares in the evidence grid (Fig. 1b). For such adviser, confidence level of 'certainly in the black urn' (score of 5B) is reported when close to 100% squares in the grid are black, indicating that probability of the coin being in the black urn is

almost certain. Conversely, lack of confidence (score of 1B) stems from weak evidence and indicates the probability of the coin being in the black urn is around chance (50%). Advice confidence would deviate if the confidence is higher (positive deviance) or lower (negative deviance) than the probability indicated by the evidence. Participants' advice deviance was significantly greater than 0 (two tailed $t$-test, t(119) = 17.08, $P < 0.00001$, Cohen's $d$ ($d$) = 1.56), as they displayed systematic overconfidence in their advice. Importantly, participants' advice deviance was affected by their current level of influence on the client, i.e., were they ignored or chosen by the client ('selection by client' variable), and was larger, i.e., more overconfident, when ignored by the client compared with trials in which the participant was the selected adviser (paired $t$-test, t(119) = 3.1, $P = 0.002$, $d = 0.28$) (Fig. 2b). Moreover, participants adjusted their advising policy dynamically: overconfidence increased in periods when they were not chosen by the client and was attenuated during periods during which they were selected by the client (Fig. 2c).

This finding demonstrated how being selected over a rival shapes the advisor's attempt to influence the client, in line with the predictions of the competitive strategy. However, our analysis so far focused only on the current level of influence on the client, measured by whether or not the adviser was selected by the client, as the force shaping our attempt to influence others, while ignoring the effect of social comparison with the rival adviser implicated in social rank theory[10–14]. A deeper understanding of

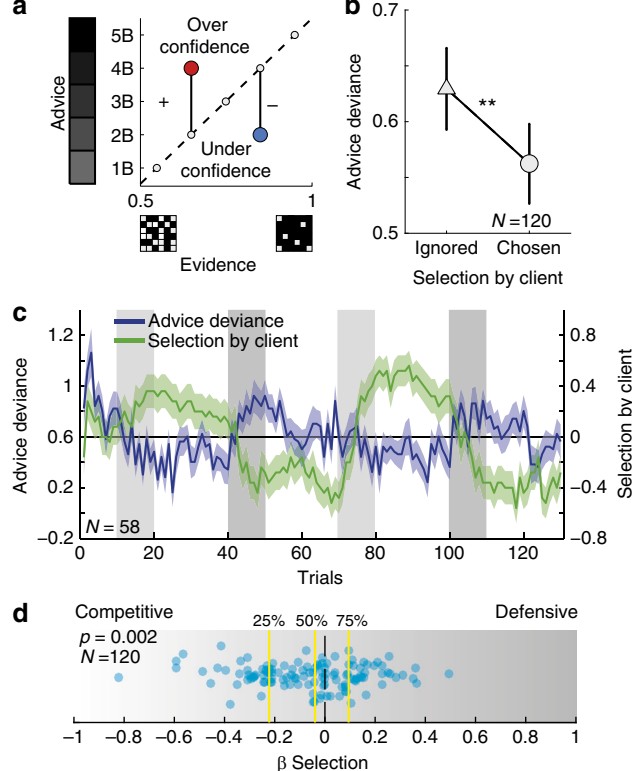

**Fig. 2** Selection by client effect on advice. **a** Deviance of advice from probabilistic evidence. Confidence is plotted against evidence grid, i.e., probability that coin is in the black urn. The dashed line represents zero-deviance policy in which confidence matches the ratio of black to white squares in the evidence grid. Overconfident advice (red circle) would lie above the dashed line and was defined as positive deviance; conversely, underconfident advice (blue circle) corresponded to negative deviance. **b** Averaged advice deviance under high vs. low selection (paired t-test, t(119) = 3.10, P = 0.002). **c** The dynamic interplay of selection and advice deviance in the online experiment. Blue line = trial-by-trial average advice deviance. Green line = average selection. Light green and blue = SEM. Light and dark grey blocks mark periods of low evidence reliability for the rival and for the participant, respectively (see Fig. S1). **d** Our best-fitting model contained a parameter governing how selection by client affect advice deviance, 'β-Selection'. Positive values of β-Selection are associated with the defensive strategy, i.e., increased advice deviance when being selected. Negative values are associated with the competitive strategy, i.e., decreased advice deviance when being chosen by the client. Despite high level of individual differences, this parameter was significantly lower than 0 across participants (Mean ± SEM = −0.08 ± 0.02, t(119) = 3.12, P < 0.002), in line with our previous analyses

the participant's advising strategy would require factoring in the interaction between the clients' selection of the advisor, an exogenous variable, and the advisor's relative merit (an endogenous variable). To disentangle the impact of the client's selection and the advisor's relative merit on the adviser's advice-giving behaviour, we employed a computational approach.

Although the client's adviser selection on each trial was made public to the participants and explicitly available for analysis, trial-by-trial variations of adviser's merit relative to the rival adviser had to be inferred to examine its effect on advice giving. We noted that participants could evaluate the prognostic value of their own and their rival's advice every time the content of the chosen urn was revealed (Fig. 3a). For example, strong advice for the black urn would have better prognostic value than a cautious

advice supporting the same choice) if the black urn contained a coin. Conversely, cautious advice for the white urn would have better prognostic value than strong advice for the white urn if the white urn was chosen but turned out to be empty. Prognostic value is calculated by multiplying advice confidence with its accuracy (correct = 1, wrong = −1) (i.e., whether the urn suggested by the advice contained a coin, see Methods for further details). On a given trial $t$, we update the relative merit variable by using the difference between the prognostic value of the participant's and the rival's advice such that:

$$\Delta PrognosticValue(t) = Confidence_{Participant}(t) \cdot Accuracy_{Participant}(t)$$
$$- Confidence_{Rival}(t) \cdot Accuracy_{Rival}(t)$$
$$(1)$$

$$RelativeMeritPE(t) = \Delta PrognosticValue(t) - RelativeMerit(t)$$
$$(2)$$

$$RelativeMerit(t+1) = RelativeMerit(t) + \gamma \cdot RelativeMeritPE(t)$$
$$(3)$$

In eq. (3) $\gamma$ is the learning rate of change to relative merit. Relative merit is positive if the participant's advice had consistently higher prognostic value compared with rival's advice. Importantly, relative merit is calculated independently from selection by the client and gives a quantitative estimate of the latent subjective process of social comparison.

To quantify whether and how this measure of relative merit could explain overconfidence behaviour, we followed a model fitting approach used in behavioural and neuroimaging studies to estimate the latent subjective processes that involve adviser's reliability, reward prediction errors and social comparisons[29,33,34]. We fitted six hierarchically nested computational models to advice deviance. Our most simple model had only one free parameter for systematic bias in advising, capturing the average (intercept) advice deviance, which stands for trait overconfidence and under-confidence (Bias Model). Our next model included the trial-by-trial selection by the client as well (Client Model). Positive values of the weight assigned to selection by client weight, would indicate that the participant followed a 'defensive' strategy, expressing higher confidence when selected by the client. Conversely, negative values for this parameter would correspond to the 'competitive' strategy. Our next model included the sign of the relative merit from eq. (3) (Merit model) and another model included both the sign of the relative merit and selection bay the client (Mixed model). Finally, we set up a model which included all the above and the interaction between selection by client and relative merit (Interaction model, eq. (4)). We also fitted models that used the sign and amplitude of relative merit (see Supplementary Materials).

$$AdviceDeviance_{Interaction}(t) = Bias + \beta_{Selection} \cdot Selection(t)$$
$$+ ... \beta_{Merit} \cdot sign(RelativeMerit(t)) + \beta_{Interaction} \cdot Selection(t) \quad (4)$$
$$\cdot sign(RelativeMerit(t))$$

After fitting all models to the advice deviance data and compensating for the number of free parameters (see Supplementary Table 1, and Supplementary Figures 2 and 3), we found that the interaction model (eq. (3)) gave the best fit to the empirical data. Our model fitting procedure estimated individual parameters for Bias, $\gamma$, $\beta_{Selection}$, $\beta_{Merit}$ and $\beta_{Interaction}$ (Supplementary Figure 3 and Supplementary Table 1). Bias parameter was significantly higher than zero across participants, verifying

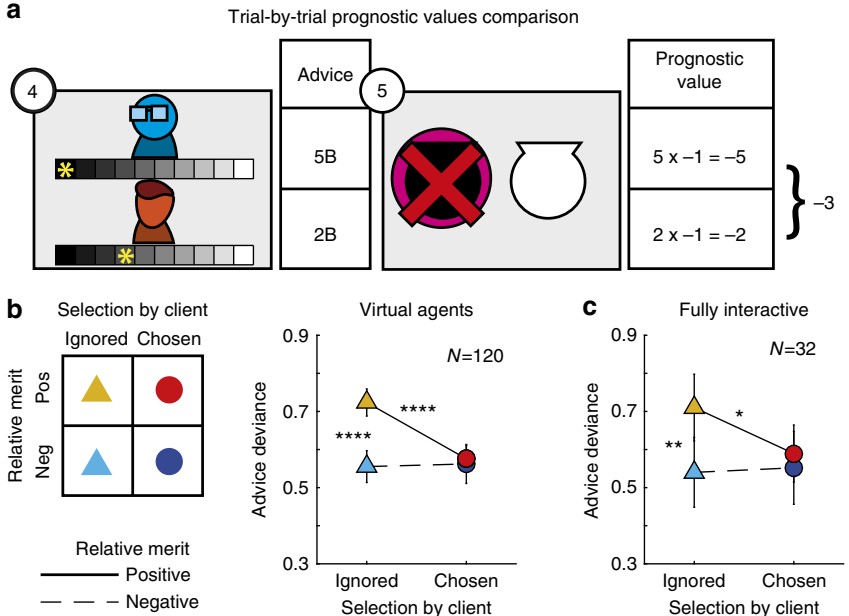

**Fig. 3** Relative merit effect on advice. **a** At the end of each trial participants could evaluate and compare their and the rival's advice prognostic value. The difference in prognostic value was accumulated to form the participants' relative merit. **b** Our best-fitting model included a free parameter for accumulation rate for differences in prognostic values, for relative merit and for the interaction between merit and client's selection. We used the individual estimated parameters to evaluate the trial by trial relative merit and divided the trials to four conditions according to the selection by client and relative merit variables. Advice deviance was highest (i.e., most overconfident) when participants' positive relative merit conflicted with the client's choice to ignore them and demonstrated a significant interaction effect. **c** The same pattern of behaviour was replicated in the fully interactive experiment, where three human participants played the roles of a client and two advisers (two-tailed paired *t*-test comparisons: * $P < 0.05$, ** $P < 0.005$, **** $P < 0.00005$, error bars indicate SEM)

our observation that participants were generally overconfident (mean ± SEM Bias: $0.67 \pm 0.05$, two-tailed *t*-test, $t(119) = 13.1$, $P < 10^{-10}$, $d = 1.29$). In addition, the selection parameter ($\beta_{\text{Selection}}$) was significantly lower than zero across participants (mean ± SEM: $-0.07 \pm 0.02$, two-tailed *t*-test, $t(119) = -3.2$, $P = 0.002$, $d = 0.28$), supporting our direct comparison in Fig. 2b, favouring the 'competitive' over 'defensive' hypothesis.

Although selection parameter ($\beta_{\text{Selection}}$) was significantly lower than 0, we observed high degrees of variability among individuals, with a positive selection parameter estimated for one third of our participants (38) (Fig. 2d). Previous works on social rank theory[8,35] suggested that negative self-esteem, i.e., seeing oneself as inferior to others and less desirable, may lead to displays of low confidence in social interactions and a greater receptivity to negative social signals. We reasoned that population variability in behavioural response to social selection and/or exclusion (as quantified here by our selection parameter) may be accounted for by individual differences in receptivity to negative social feedback. To test this hypothesis, we went back to our pool of participants and were successful at recruiting $N = 69$ of them to complete the Fear of Negative Evaluation (FNE) questionnaire[8,36]. We found that FNE score correlated with the participants' model estimate of selection parameter, as participants with higher FNE score, i.e., more negative self-perception, were more likely to follow the defensive strategy ($N = 69$, Pearson's correlation $R = 0.25$, $R^2 = 0.06$, $P = 0.036$). By linking previous research on social rank and our participants' behaviour, this latter finding provided evidence of external validity for our computational model.

Estimated model parameters associated with relative merit, and the interaction term between relative merit and selection by client were harder to interpret directly. Even greater individual differences were observed for these parameters. When averaged across participants, neither parameter was significantly different

from zero (two-tailed *t*-test, $t(119) = < 0.7$, $P > 0.45$), with individuals varying in the effect of relative merit and interaction (see Supplementary Table 1). However, model selection and comparison showed beyond doubt that the interaction parameter did provide the model with better power to capture behaviour. To examine the contribution of all parameters to the explanation of the behavioural data, we used the sign of the relative merit estimated by the interaction model using the fitted relative merit learning rate $\gamma$ (eq. (3)), to label the trials as 'positive' vs. 'negative' relative merit. Each trial was also categorized according to the selection by the client as 'ignored' or 'chosen'. We then examined advice deviance across the resulting $2 \times 2$ combinations of relative merit (positive vs. negative) and selection by client (selected vs. ignored) using a repeated-measures analysis of variance (ANOVA). We found a significant effect of relative merit (F $(1,358) = 14.4$ $P = 0.0002$, $\eta^2_{Partial} = 0.14$), a significant effect of selection by client (F$(1,358) = 8.1$, $P = 0.005$, $\eta^2_{Partial} = 0.04$) and a significant interaction effect (F$(1,358) = 17.95$, $P < 0.0001$, $\eta^2_{Partial} = 0.14$) on advice deviance, as model fitting procedure had suggested (Fig. 3b and see Supplementary Figure 5 for these results for each cohort (online, lab and scanner) separately). Importantly, this analysis elucidated the nature of the interaction by showing that participants expressed significantly greater confidence in their advice on trials in which they assumed they had done better than their rival (i.e., positive relative merit) but nonetheless had been (perhaps unexpectedly) ignored by the client. To put it metaphorically, advisers shouted most loudly when they had reason to believe that their merits had been overlooked. We used a simulation to examine whether such pattern could be recovered by one of our competing models (bias, client, merit, mixture and amplitude, as described above)[37] and found that only the interaction model (eq. (4)) could reproduce this pattern of results (Supplementary Figure 4).

The behaviour we observed—namely increasing advice confidence when ignored by the client while having positive relative merit (Fig. 3)—could potentially be a response to the specific manner in which our algorithms controlled the client and rival adviser behaviour (which they did by following Bayarri and DeGroot's assumptions[3]). To confirm the validity and generality of our results, we ran a fully interactive experiment in a fourth cohort of participants, in which both the advisers and the client were human participants (see Methods for details) and no agent's behaviour was under experimenter control. The scenario was the same as before, but now involved three participants engaged in a multiplayer game played on three computers in three adjacent cubicles connected via the internet. We applied the same analysis and model fitting, and estimation of relative merit and selection by client effects to the advisers' behaviour in this fully interactive experiment. The results replicated the main results from the virtual agents' experiment: advice deviance was highest when the adviser was ignored but her relative merit was positive (Fig. 3c; mixed-effects ANOVA; a significant interaction effect (F(1,96) = 5.05, $P = 0.03$, $\eta^2_{Partial} = 0.14$, and a significant effect of relative merit, F(1,96) = 4.43, $P = 0.04$, $\eta^2_{Partial} = 0.125$). In addition, we did not observe any significant discrepancies between live client's behaviour and virtual client behaviour, when comparing the real clients and algorithm simulation proportion of choosing adviser 1 (paired $t$-test, t(15) = 1.45, $P = 0.17$) (Supplementary Figure 6).

To further examine the relation between the virtual and live experiments, we fitted data from all experimental cohorts, within a mixed-effect ANOVA with relative merit (positive/negative), selection by client (chosen/ignored) and agents (live/virtual) as main factors and subjects as random-effect factor nested within the agents factor. We did not observe any effect of agents on advice, neither in the main effects nor the interaction. Overall, we observed a significant main effect of relative merit (F(1,442) = 15.85 $P = 0.0001$, $\eta^2_{Partial} = 0.096$), a significant main effect of selection by client (F(1, 442) = 6.17, $P = 0.014$, $\eta^2_{Partial} = 0.04$) and a significant interaction between the two factors (F(1, 442) = 9.28, $P = 0.002$, $\eta^2_{Partial} = 0.06$).

The results from the fully interactive experiment provided an additional replication of our main results, demonstrating the robustness of the findings and strengthening the ecological validity of the paradigm in generalizing the findings to interactive human behaviour.

**Neural correlates of selection by client.** Having established the effects of selection by client and relative merit on advising behaviour, we used the same paradigm to examine the neural correlates of the attempt to influence others. Participants played the role of adviser in the advice-giving game with the computer algorithm controlling the client and rival adviser responses, while lying in an MRI scanner. Each of the five stages of a trial in the advice-giving game was treated as an event in a fast event-related design (see Methods). Using the computational model-based analysis described above, we labelled trials according to their relative merit (positive vs. negative) and selection by client (ignored vs. chosen). The resulting $2 \times 2$ combination of conditions allowed us to examine the changes in the blood-oxygenation-level-dependent (BOLD) signal that correlated with the main effects of selection and its interaction with relative merit. Activity in the rTPJ ($P < 0.001$, family-wise error (FWE) cluster size-corrected $P < 0.05$) (Fig. 4 and Supplementary table 2) displayed the main effect of selection by client (Ignored > Selected): BOLD signal in this region was higher when the participant was ignored (vs. selected) by the client (regardless of the sign of relative merit) during the observation of evidence. This finding is in line with previous studies showing increased activity in the rTPJ when

others' actions did not match the participant's predictions[38], and when inferring others' intentions from their actions[20].

When evaluating the interaction model parameters, we observed that trait negative self-perception, as assessed by the FNE score[8,36], was associated with tendency to follow the defensive strategy of advice giving. Based on that observation, we hypothesized that FNE scores may predict the brain activity associated with Selection by the Client. Twenty-eight of the participants that took part in the neuroimaging experiment also completed the FNE questionnaire. Consistent with our hypothesis, we found that the rTPJ response to selection by the client (ignored > chosen, Fig. 4) was correlated with participants' individual FNE score (Fig. 4c) ($N = 28$, Pearson's correlation $R = 0.65$, $R^2 = 0.39$, $P = 0.0004$). Sensitivity of rTPJ to selection by client (ignored > chosen) increased with participant's inferior self-perception, captured by the FNE scores.

To further examine the effect of selection by client, we used a 'client selection switch' predictor as a regressor in a whole brain general linear model (GLM) analysis. The 'client selection switch' predictor was set to +1 on trials when the client switched from the rival adviser to the participant, to −1 when switching from the participant to the rival adviser, and to zero when the client did not switch adviser. We found that the VS activity was modulated by the 'client selection switch' variable during the evidence observation stage. VS activity increased when the client switched from the rival adviser to the participant and decreased when client switched from the participant to the rival (Fig. 4d, $P < 0.001$, FWE cluster size-corrected $P < 0.05$, see Supplementary table 3 and Supplementary figure S7 for uncorrected map). The 'client selection switch' variable reflects the changes in the client selection, similar to the way prediction errors reflect changes in expected reward[30], and are in line with reports of increased VS responses when selected by a client[31]. To summarize, trial-by-trial 'selection by client' was tracked by the rTPJ activity, while client selection switches were tracked in the VS.

**Neural correlates of relative merit.** At the outcome stage, participants had all the information necessary to compare the prognostic value of their own and the rival's advice (Fig. 3, eq. (1)), and to update their relative merit variable by computing the relative merit prediction error (PE) (eq. (2 and 3)). Using the model parameters evaluated for each individual, we estimated the trial-by-trial relative merit and relative merit PE variables values, and used these as predictors in a whole brain parametric modulations analysis (see Methods). We found that during the outcome stage, activity in mPFC tracked the trial-by-trial changes in relative merit PE (Fig. 5a, $P < 0.001$, FWE cluster size-corrected $P < 0.05$, see Supplementary table 4 and Supplementary figure 5 for uncorrected map). The findings reported here support the role of mPFC in social comparison with the rival adviser, tracking trial by trial the relative performance used to update relative merit. Moreover they are consistent with involvement of mPFC in evaluating and inferring other agents' traits such as reliability[29] and accuracy[26,27] and beliefs[39], and in evaluating one's rank in relation to others[24].

Using the relative merit PE in a whole brain parametric modulation analysis during the Appraisal stage (Fig. 1) we found that the VS tracked the trial by trial fluctuations in relative merit PE (Fig. 5b, $P < 0.001$, FEW cluster size-corrected $P < 0.05$, see Table ST5, and Figure S8 for uncorrected map showing bilateral VS activity). VS responses to prediction errors in social comparison are in line with previous studies showing similar VS responses to interpersonal prediction errors[40], and to relative performance[26].

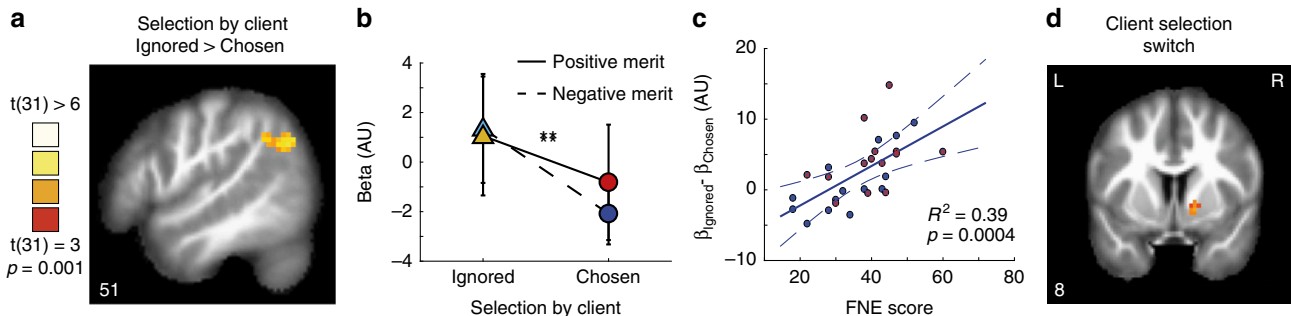

**Fig. 4** Encoding of selection by client and client selection switch. **a** During the observation of evidence stage, activity in right TPJ (coordinates: [51,−58,35]) was higher on ignored vs. chosen trials ($P < 0.001$, FWE cluster size-corrected $P < 0.05$). **b** Analysis of beta estimates from the right TPJ showed a significant selection by client effect ($N = 32$, $F(1,94) = 11.4$, $P = 0.002$). Error bars indicate 2 SEM. **c** Client selection effect on activity in the right TPJ was correlated with the participants' fear of negative evaluation (FNE) score: differences in rTPJ activity between ignored and chosen trials was greater for participants with high FNE score ($N = 28$, as we obtained FNE scores from a subset of the participants. Red cross indicates female participants). Regression line is computed from the population-level estimate of the FNE scores on selection by client effect in the rTPJ. Dashed lines indicate 2 SEs. **d** At the evidence stage activity in VS followed client selection switches in adviser selection, i.e., higher when the client switched from the other adviser to you and lower when switched from you to the other adviser ($P < 0.005$, FWE cluster size-corrected $P < 0.05$, see Figure S8 for uncorrected map

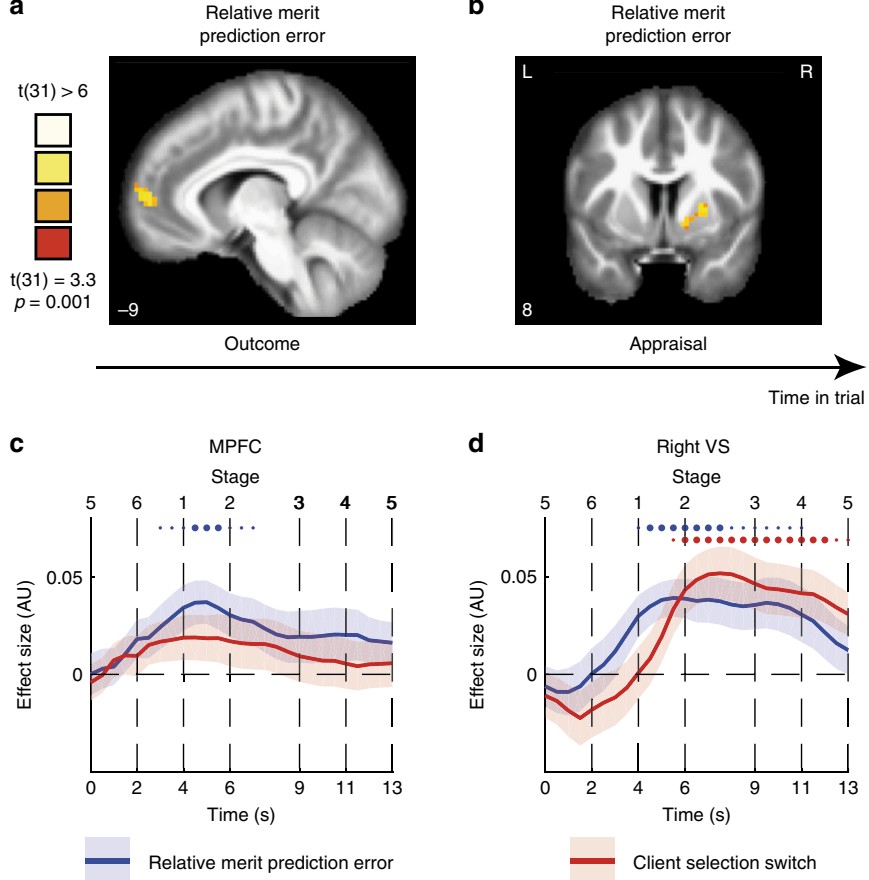

**Fig. 5** Encoding of relative merit prediction errors. During the outcome stage, a whole brain analysis showed that activity in mPFC was positively modulated by the trial-by-trial relative merit prediction error, i.e. the differences between prognostic value comparison and previous reltive merit (Fig. 2d, eq. (1)) ($P < 0.001$, FWE cluster size-corrected $P < 0.05$, see Figure S7 for uncorrected map). **b** During the Appraisal stage, activity in the VS followed the relative merit prediction error ($P < 0.001$, FWE cluster size-corrected $P < 0.05$, see Figure S8 for uncorrected map). **c** Time course of the effects of relative merit prediction error (blue) and client selection switches (red) from the mPFC (MNI coordinates [x,y,z]: [−9,56,5]). Time courses are presented across all stages of a trial, from the showdown stage (5) of a preceding trial to the showdown stage (5) of the current trial. Thick lines indicate mean effect size, the shaddows indicate the SEM, small dots indicate $P < 0.05$ in these time points, and big dots indicate $P < 0.005$. These time courses demonstrate the temporal order of coding of relative merit and selection by client in VS. Thick lines: mean effect size. Shadows: SEM. **d** Time courses sampled from the right VS (coordinates: [18,8,−10], defined by Neurosynth), demonstrating the temporal order of coding of relative merit PE and client selection switch

To further explore the temporal dynamics of tracking relative merit PE and client selection switches, we used a region of interest (ROI) approach to examine the time courses from the mPFC and the right VS (and left VS, see Fig. S8). These ROIs were defined using spheres around the coordinates indicated by NeuroSynth[41] forward inference maps for the term 'Ventral Striatum' and around the peak activation in the mPFC (see Methods). We followed the steps used in previous studies[29] to examine time courses of trials containing multiple jittered stages, aligning each trial stage timings to the stage's mean time. We then performed a GLM on each time point across trials, in each participant, regressing the trial-by-trial relative merit PE and the client selection switch variables from the BOLD signal and evaluating group effects (Fig. 5c, d). We found that the mPFC response to relative merit PE, but not for client selection switch, increased after the outcome stage (two-tailed $t$-test of $\beta$-coefficients, t(31) > 2.8, $P < 0.005$, see dots in Fig. 5c). In the VS, responses to relative merit PE reached significance (two-tailed t-test of $\beta$-coefficients, t (31) > 2.8, $P < 0.005$, see dots in Fig. 5d) before the responses to client selection switch.

To summarize, the interaction between relative merit (vs. rival adviser) and current level of influence on the client shape advice confidence. Across four experiments, confidence was highest when recent history indicated that a given adviser had performed better than her rival, but the client still chose to listen to the rival. Individual differences adherence to this strategy were associated with personality trait of lower self-perception, captured by high FNE scores. Brain activity in the right TPJ covaried with selection by the client and this effect was enhanced in participants with high FNE score. Switches in client's choice of adviser were tracked by activity in the VS. Fluctuations in relative merit prediction errors were correlated with mPFC and VS response.

## Discussion

We set out to study how humans attempt to influence others and found a pattern of behavioural results consistent across multiple cohorts of participants playing the advice-giving game with either virtual or real confederates in web-based and laboratory-based experiments (respectively). Advice giving behaviour was driven by the interaction between two factors: the current level of influence over the client and the relative (i.e., social comparative) merit over the rival. Most prominently, when participants' relative merit was positive (i.e., they were doing better than their rival), they followed what we called the 'competitive' strategy[3] by expressing higher confidence when ignored by the client and lower confidence when selected by the client. Inter-individual variability in participants' behaviour was captured by a personalized quantitative combination of the impact of these two factors on advice confidence placing each participant's strategy along a spectrum ranging from competitive[3] to defensive[9]. This quantitative profile was predictive of the participant's self-reported FNE score[8], thus grounding our paradigm firmly in previous research on self-esteem. Using fMRI, we found that the mPFC and VS tracked the changes in the participants' inferred merit relative to the rival adviser (relative merit PE). The rTPJ tracked the client selection of adviser on each trial, with increased activity on trials in which the participant was ignored. Switches in client selection (from the participant to the rival adviser and vice versa) were tracked by VS activity. The temporal dynamics of the computations tracked by the BOLD signal in the VS were consistent with the unfolding of events across the trial, with relative merit PE tracked before selection by client.

Our study goes beyond previous investigations of the neurocognitive basis of social influence in a number of key aspects. First, it put the participants (advisers) in a position to influence

others' (client) choices. Almost all previous studies of social influence invariably concentrated on how one reacts to attempts by others to shape their behaviour. For example, Campbell–Meiklejohn et al.[42], and Izuma and Adolphs[43] investigated how our preferences change when we observe others' opinion about objects of variable value. In these studies, the social influence was predefined by the experimenter and the participants' behaviour could not change this influence. Zink et al.[25] and Lignuel et al.[34] investigated how people infer their status in a dominance hierarchy from competitive and/or cooperative interaction with others. The results from Ligneul et al.[34] showed that mPFC was involved in the dynamic representation of one's status in a group's hierarchy and in the manner this information is used when choosing (or avoiding) opponents in subsequent competitive or cooperative encounters. However, these studies focused on how the participant reacted to others' attempts to shape the participant's behaviour and therefore did not examine the ways participants might attempt to affect their opponent nor how participants may try to establish their own dominance so that trouble-making opponents would avoid them. Although Mobbs et al.[31] studied advice giving behaviour directly—and showed that VS activity increased when the participant's advice was taken and beneficial, and the mPFC activity following the advice consequences—their scenario was not interactive as the client's responses and participants' accuracy were set by the experimenter and advisers could not use any strategic signal (such as confidence) to influence the client. Here we studied advisers' strategic behaviour as they tried to influence a client in an interactive context similar to other game theoretic treatments of social influence such as the inspector[20,44] and the trust game[45].

The second way in which our study goes beyond previous investigations of the neurocognitive basis of social influence is in the social relationships between the confederates. Although previous research studied human social interactions with more than one confederate, none placed their participants in a context involving dissimilar (asymmetric) social relationships to the confederates. In many studies, social interactions involved tracking another person's reliability or intentions[20,29], or tracking multiple agents all having a similar role[26–28,42]. Other studies have examined how people evaluated group behaviour, where the groups consisted of agents with similar incentives and roles and the participants engaged in various tasks such as competitive bidding[46], reaching consensus[47], inferring one's hierarchical rank[24,34] or tracking the group's preferences[32]. However, in our task—as is the case in many real-life scenarios—the participant had to track two distinct types of relationships: a competitive one with a rival adviser and a hierarchical one with a client whose appraisal they sought. This novel configuration of asymmetric social relationships allowed us to disentangle the separate contributions of the elements of the 'social brain' system[19,22], namely that of rTPJ and mPFC in influencing others. These contributions consisted of an internally and an externally driven processes. The internal inference process was tuned to social comparison and the computation of one's relative merit in the mPFC[24,26,39]. The external process was tuned to the evaluation of social outcomes arising from others' behaviour in relation to the self, in our case the selection by client, in the rTPJ. Finally, one may interpret the selection by client as an external event to which the participant's attention is oriented at the beginning of every trial, in contrast, for example, to the latent process of social comparison tracked in the mPFC. The rTPJ activity elicited by external social events would therefore be in line with previous findings about the involvement of rTPJ in the orienting of attention to salient external events[48,49]. This distinction is also in line with previous results examining strategic behaviour in two-person games[20], where mPFC was associated with internal inference processes and

the rTPJ with evaluating external signals about the behaviour of the other.

Our findings also provide evidence for the involvement of the VS in social behaviour. Changes in both relative merit and selection by client affected VS BOLD activity, a neural structure implicated in valuation of motivational factors in decision making. This observation is in line with the combined interactive impact of these two factors on advising behaviour. Previous studies have shown that striatal neural activity may encode multiple social attributes such as reputation[50], selection[25], appraisal[31] and vicarious reward[51], and also in tracking other's behaviour and interpersonal prediction error[40]. Here striatal activity was correlated with the different computational components of the interactive scenario as they emerged during the course of the interaction: social comparative merit and social appraisal. This finding supports the notion that the VS has a domain-general and dynamic role in valuation of various events in our environment as they occur[30,52].

In addition, our results indicate the importance of individual differences in social self-perception on strategic social behaviour. Computational analysis of behaviour revealed considerable variations in participants' advising strategies, with each participant's behaviour falling on a continuum from a pure 'defensive'[9] to a pure 'competitive'[3] strategy. These variations were captured by a personalized quantitative combination of the impact of relative merit and selection by client on advice confidence and were consistent with social rank theory: participants with negative self-perception, scoring high on FNE, were more likely to follow the defensive strategy[8]. In the neuroimaging experiment, participants with higher scores of negative self-perception displayed increased response in the rTPJ to being ignored by the client. These findings provide converging evidence linking management of social influence to negative self-perception. Our cognitive framework should aid future research on the social basis of mental health disorders such as depression. Deterioration of self-esteem and retreating from social engagement are two of the earliest and most debilitating hallmarks of depression[8]. The laboratory model described here offers a uniquely appropriate ecologically valid tool for measuring these social cognitive characteristics of depression.

Finally, our results demonstrate how people use confidence reports as a persuasive signal in a strategic manner. Such use of confidence reports is in line with the literature on persuasion and information sharing[3,4,53]. However, numerous studies have used similar confidence reports to study the process of metacognition, the internal process of evaluating one's precepts and decisions[54,55]. Our findings demonstrate how, depending on the social context, confidence reports can depart from simply describing uncertainty about sensory information or decision variables. The findings underscore the importance of taking extra care about the framing (e.g. experimental instructions) and phrasing of how to ask participants to report their confidence in psychological and neuroscientific investigations. Our findings support the view of metacognition as a multi-layered process[56], in which expressing one's confidence depends not only on uncertainty about low level sensory processing or decision processes aiming to maximize reward, but also on other systematic, non-trivial sources of variance such as social comparison, closeness and friendship[57,58], and social expectation[59].

People with a more accurate opinion are often more confident. However, the converse is not necessarily true: being more confident is not necessarily predictive of accuracy. Many keen observers of human condition (e.g., Bertrand Russell, W. B. Yeats and William Shakespeare, to name but a few) have complained that people who know a lot are fraught with self-doubt, whereas

the ignorant are passionately confident. Our behavioural and neurobiological results suggest that passionate overconfidence of the underdog could be better understood as a sensible recourse to the competitive strategy designed to gain higher social influence, a behaviour supported by the brain's social and valuation systems. The philosophers' sad lamentation therefore highlights the importance of the social comparison processes, in moderating the competitive behaviour of the ignorant.

## Methods

**Participants.** We recruited four cohorts of participants for this study. All participants provided informed consent, and received monetary compensation. The study was approved by the research ethics committee at University College London (UCL). Following a pilot experiment involving 20 participants, we estimated our effect size to be around 0.5. As our experiment follows a within-participants design, we decided to recruit 60 participants for the online experiment and 30 participants for the longer lab based experiment. We recruited 60 participants for the online experiment using Amazon M-Turk. Two online participants were excluded from analysis, as they did not use the full confidence scale. Online participants included 31 males (ages mean ± SD 33.7 ± 9.6) and 27 females (ages 36 ± 8.5). We recruited 30 participants for a lab based experiment, in which participants carried the experiment on computers in the Psychology department building. One participant was excluded from analysis as she used only one advice level. Lab participants included 13 males (ages 26.5 ± 6) and 16 females (ages 26.2 ± 6). Finally, 34 participants were recruited for the neuroimaging part of the experiment. These corresponded to the expected effect size of activity in previous social cognition neuroimaging studies, e.g., see refs [25,31,43]. Of these, two participants were excluded from analysis due to head movements and data corruption. Neuroimaging participants therefore included 18 males (ages 24.7 ± 6.6) and 14 females (ages 23.78 ± 4.6). Finally, in the fully interactive experiment we collected data from 19 triplets (57 participants, 24 males aged 25.2 ± 4.76, and 33 females aged 21.2 ± 2.25) and excluded 3 triplets from analysis. These included two triplets in whom advisers used only the highest confidence levels (4 and 5) and one triplet in whom the client chose only one adviser throughout the experiment. We therefore analysed the data from 32 advisers in the live interaction experiment.

**Client and rival adviser algorithms.** All participants in the main experiment played the role of an adviser, whereas the client and the other adviser were played by computer algorithm. The other adviser's advice were calculated on each trial according to the probability of the coin being in the black urn (between 0–1), plus noise (~ $N$ (0,0.08)), to range between [5W 5B], just like the participants' advice. After outcome is revealed on each trial, both advice's prognostic value is calculated by multiplying the confidence level (1–5) by accuracy (indicating the correct coin location, i.e., W for white or B for Black urn, 1 = correct, − 1 = incorrect, eq. (1)).

The client's choice of adviser was determined by assigning an influence weight to each adviser, updating the weights after each outcome and choosing the adviser with the higher weight in the next trial. The weights summed to 10 and were set to be 5 for each adviser in the beginning of the experiment. To update the client influence weights, we used prognostic value of advice (PA), which were derived from advice according to the following rule: when the black urn is suggested then confidence is between [− 5, − 1] and PA is calculated by confidence + 6 to range between [1, 5]. When the white urn is suggested then confidence is between [1, 5] and PA is calculated by confidence + 5 to range between [6, 10]. We used the notation $PA_P$ for to refer to prognostic value of the participant's advice and $PA_O$ for prognostic value of the other's (rival) advice. The weights were updated after each trial according to the last trials' prognostic values, following a rule similar to the one used by Bayarri and DeGroot[3]:

$$w_P(t+1) = 10 \cdot \frac{w_P(t) \cdot PA_P(t)^2}{w_P(t) \cdot PA_P(t)^2 + w_O(t) \cdot PA_O(t)^2} \qquad (5)$$

Where $w_P$ is the influence weight assigned to the participant. The other adviser's influence weight was defined as $w_O(t+1) = 10 - w_P(t+1)$. It is noteworthy that when the influence weight of one adviser increases, the other adviser's influence decreases in the same amount. When both advisers give the same advice the influence weights remain the same.

**Experimental procedure.** We carried out the main experiment on three different platforms: online, in the lab and in the scanner. The main experimental design features were the same across these experiments with a number of minor differences in implementation.

In the online experiment, participants were recruited using Amazon M-Turk. These participants carried out the experimental task online on their own computers using the mouse to input their confidence rating. They received a fixed monetary compensation and were promised a bonus if the client selected them on more than 100 trials. The online experiment had 130 trials. Evidence stage lasted 500 ms and all other stages of the trial were self-paced. Advice giving stage ended when

confidence was reported. After the outcome was displayed, participant proceeded to the next trial by pressing a 'Next' button.

Lab-based participants were invited to the lab in groups of three and were told that they are about to play an adviser game together, and that the roles of two advisers and client will be assigned randomly at the beginning of the experiment. The participants were then seated in isolated individual cubicles. Unbeknownst to the participants, all three players were assigned to the adviser role and played against a virtual client and a virtual rival adviser. Lab-based participants received a fixed monetary compensation, and did not have any further incentive. Lab-based experiment consisted of four blocks of 70 trials. Advice giving stage was self-paced. All other stages lasted a predefined length of time: appraisal stage lasted 1.5 s, evidence stage lasted 500 ms, showdown stage lasted 2 s and outcome stage lasted 2 s.

In the neuroimaging experiment, participants arrived at the scanner unit and met two confederates and were given the same cover story as the lab-based participants. Participants were then put in the scanner, and were instructed on the use of response boxes for inputting their confidence ratings: left-hand response box shifted the rating towards the black urn, right-hand response box shifted them towards the white urn. Participants received a fixed monetary compensation. In this experiment, intervals between stages were set to 1.5 s plus a jittered interval sampled from a Poisson distribution (range 0–3 s; mean 1.5 s). Neuroimaging experiment consisted of four blocks with 60 trials. Advice giving stage was self-paced. All other stages lasted between 1.5 and 4.5 s. In addition, four intervals of 8 s rest were randomly dispersed between trials of each run.

In addition to the main experiment, we ran a fully interactive experiment in which all roles in the task, advisers and client, were played by human participants. This experiment was carried in a similar manner to the lab-based main experiment, with participants arriving in groups of three, and carrying the experiment on computers in separate cubicles where they were randomly assigned to the roles of advisers or client. Participants received a fixed monetary compensation and did not get any further incentive. The experiment consisted of 130 trials. The pace of the experiment depended on the interaction between the participants. Advisers waited for the client to choose one of them and then the client waited for both advisers to express their advice. See analysis of live and virtual clients' behaviour in Supplementary Materials figure S6.

**Selective manipulation of advice quality**. In our early pilot sessions, we noticed that sometimes the virtual client did not shift between participants, practically ignoring one of them throughout the experiment. This happened because advisers tended to be very similarly calibrated with the evidence. As we intended to use a within-participants design to compare advice on periods in which the adviser is chosen and periods in which he is ignored by the client, we needed a way to manipulate the probability of switching between selected and ignored status. Therefore, in restricted periods of a block, we introduced some noise to one of the two advisers' evidence such that the ratio of black and white squares in the grid became a poor predictor of the reward location. This procedure went as follows: if on a specific trial the probability of the coin being in the black urn was 0.75, the grid would normally include 75 black and 25 white squares (Figure S1). On a noisy trial with similar probability of 0.75, this composition would change to 55 black squares and 45 white squares, akin to a reduction in contrast by 20 squares. In all noisy trials contrasts were reduced by 20 squares in a similar manner (Figure S1). The procedure ensured that one advisor's advice accuracy was systematically inferior to the other one for a number of consecutive trials thus increasing the probability that the virtual client would shift to selecting the other adviser.

To ensure that the evidence quality manipulation was not the sole driver of changes in accuracy and difficulty throughout the task, the order of the evidence displayed to the participants in each trial (the ratio between black and white squares in the grid) was randomized for each participant. In addition, the coin location on each trial was randomly generated according to the evidence (evidence only implied the probability of the coin location). This meant that each participant experienced individual periods of high/low accuracy, which were not time locked to the evidence quality manipulation.

**Model fitting procedure**. We used models with increased complexity to explain advice deviance reported by the participants. The most elaborated model is the interaction model, described in eq. (4), which assumes that advice deviance is affected by: systematic bias in confidence (which is in fact the intercept or alpha parameter), current selection by client (ignored/chosen), relative merit tracked by comparison with rival (eq. (3)), and the interaction between relative merit and selection by client. Simpler models included: a mixture model excluding the interaction parameter, a relative merit model excluding all selection parameters, a selection model excluding all relative merit parameters and a bias model excluding all relative merit and selection by client parameters. An additional model was also tested that was identical to the 'Interaction' model but used the magnitude and sign of relative merit instead of only the sign of relative merit.

We fitted all models to individual advice deviance. We used a cost function, $L$ ($M$) to estimate a given model $M$ fit to the data. The cost function compared the advice deviance estimated with the model $M$ ($AdviceDeviance_M(t)$) and the actual advice deviance observed in behaviour on each trial ($AdviceDeviance_{Data}(t)$):

$$L(M) = -\sum_{t=1..T} \log\left(\frac{1}{1+abs(AdviceDeviance_M(t) - AdviceDeviance_{Data}(t))}\right) \quad (6)$$

Similar to log likelihood cost function, the ratio inside the log is close to one when the estimation is close to the data and it gets closer to zero when the distance between estimation and data increases. Therefore, lower values of the cost function indicate better fit of the model to the data. We used a Markov-Chain Monte Carlo (MCMC) Metropolis-Hastings algorithm for model fitting and estimation for each participant[60–62]. For model comparisons we calculated individual Deviance Information Criterion[61], which uses the distribution of likelihood obtained and penalizes for increased number of parameters (Supplementary Figure S2). We used in house Matlab code and the MCMC toolbox for Matlab by Marko Laine (http://helios.fmi.fi/~lainema/mcmc/#sec-4).

Parameter estimation was done individually by integrating the marginal distribution of the parameter values, uncovered using the Markov process chain[62]. The learning rate parameters estimated using the Interaction model were used in the aggregated analysis to determine the trial by trial relative merit and separate trials to positive and negative relative merit. The mean parameter estimations for all models are reported in table ST1 in the supplementary materials.

**MRI data acquisition**. Structural and functional MRI data were acquired using Siemens Avanto 1.5 T scanner equipped with a 32-channel head coil at the Birkbeck-UCL Centre for Neuroimaging. The echoplanar image) sequence was acquired in an ascending manner, at an oblique angle (≈ 30°) to the AC–PC line to decrease the impact of susceptibility artefact in the orbitofrontal cortex[63] with the following acquisition parameters: volumes, 44 2 mm slices, 1 mm slice gap; echo time = 50 ms; repetition time = 3,740 ms; flip angle = 90°; field of view = 192 mm; matrix size = 64 × 64. As the time for each block was dependent on the participants' reaction time, overall functional blocks changed in length and approximately 250 volumes were acquired in about 15 min and 40 s. A structural image was collected for each participant using MP-RAGE (TR = 2730 ms, TE = 3.57 ms, voxel size = 1 mm³, 176 slices). In addition, a gradient field mapping was acquired for each participant.

**fMRI data analysis**. Imaging data were analysed using Matlab (R2013b) and Statistical Parametric Mapping software (SPM12; Wellcome Trust Centre for Neuroimaging, London, UK). Images were corrected for field inhomogeneity and corrected for head motion. They were subsequently realigned, coregistered, normalized to the Montreal Neurological Institute template, spatially smoothed (8 mm FWHM Gaussian kernel), and high filtered (128 s) following SPM12 standard preprocessing procedures.

We carried two complementary data analyses. Using the computational behaviour analysis, we labelled trials according to selection by client (chosen/ignored) and relative merit (positive/negative). We examined the effect of selection by client, relative merit and their interaction on brain activity. We used individual level GLM with stick predictors at the onset of each stage (appraisal, evidence, advice report, other advice display and outcome) and additional boxcar predictor 1.5 s long ending at the time of advice report confirmation, capturing the motor button presses. We ran GLMs in which the condition labels (selection by client (chosen/ignored) and relative merit (positive/negative)) applied to the outcome stage, appraisal stage and evidence stage separately, to overcome the problem of correlation between predictors of interest, as the order of our stages was fixed. In addition, we ran a whole brain parametric modulation analysis with a 'client selection switch' predictor, set to 1 when the client switched from the rival adviser to the participant, −1 when switching from the participant to the rival adviser and 0 when the client made the same choice as before.

In a separate analysis we examined how the trial-by-trial relative merit PE-modulated brain activity. We used a set of variables as parametric modulators of activity, which included the relative merit PE as the regressor of interest, and aggregated relative merit, the trial-by-trial prognostic value comparison, client's reward prediction error and unsigned participant's advice as regressors of no interest. We used different GLMs to estimate the effect of this set of parameter modulations on the outcome stage, appraisal stage and evidence stage separately.

In ROI analysis we examined event related effects in specific brain region in different stages within an advice giving trial. Individual β-maps were estimated and sampled within ROIs using MarsBar SPM toolbox. We used NeuroSynth[41] defined ROIs for the left and right VS. We selected the peak of the VS reverse inference map, made from 310 studies. We used a 12 mm sphere around the left and right peak activity as ROI using MarsBar SPM toolbox (MNI coordinates [− 12, 8, − 8], [10, 6, − 8], z > 22).

To examine the time course of the changes in brain activity in the regions of interest, we followed previous studies[29] and exploited the time jitters to disentangle the brain activity corresponding to different cognitive processes of interest. We separated each subject's time series sampled from the VS into each trial and resampled each trial to a duration of 15 s, aligned according to the trial stages (Fig. 1): previous trial's showdown(stage 5) at time 0, previous trial's outcome (stage 6) at time 2 s, appraisal (stage 1) at time 4 s, evidence (stage 2) at time 6 s, confidence report (stages 3–4) at time 9 s–11 s and current trial's showdown (stage

5) at time 13 s (these timings were the mean timings across all trials in all subjects). The resampling resolution was 100 ms. This temporal realignment allowed the observation of signal throughout the trial while taking advantage of the random jitter and fast event-related design. We then performed a GLM at each time point across trials in each subject. We had one regressor for 'relative merit PE' and another for 'client selection switch'. We then calculated the mean of the effect across subjects at each time point and their SEs.

**Questionnaires**. As follow up for our experiment we sent participants FNE questionnaire[36] to be filled online, to test the link between trait selection perception and social behaviour, as predicted by social rank theory[35]. The questionnaire was sent six months after the main experiment, to make sure there is no effect of the questionnaires on the performance in the task and vice versa. Participants were paid for completing the questionnaires. 69 of our original 120 participants filled the questionnaire, 29 from the fMRI cohort, 15 from the lab cohort and 25 from the online cohort.

**Data Availability**. The behavioural data that support the findings of this study are available from figshare (https://doi.org/10.6084/m9.figshare.5414350.v1)[64]. The statistical parametric maps from the neuroimaging part of the experiment are available from NeuroVault: http://neurovault.org/collections/2204/.

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

## Acknowledgements

U.H. and B.B. are supported by the European Research Council (NeuroCoDec 309865). S.P. was supported by a Marie Sklodowska-Curie Individual European Fellowship (PIEF-GA-2012 Grant 328822) and is currently supported by an ATIP-Avenir grant (R16069JS) and a Collaborative Research in Computational Neuroscience ANR-NSF grant (ANR-16-NEUC-0004). C.D.F. is supported by the Arts and Humanities Research Council (grant AH/M005933/1).

## Author contributions

U.H., B.B. and C.D.F. designed the experiment. U.H. programmed the task. U.H., S.B. and C.O. collected the data. U.H. and S.P. carried out the data analysis. U.H., B.B., S.P. and C.D.F. wrote the manuscript.

## Additional information

**Competing interests:** The authors declare no competing financial interests.

