## [Peer Review File · Nature Communications]

Reviewers' comments:

Reviewer #1 (Remarks to the Author):

Hertz et al. have examined how advisors strategically modulate their advice in order to influence their client. The authors have identified key behavioral factors that guide the advisors' decision-making and revealed the neural correlates of the factors: relative merit of their advice (compared with a rival advisor's) encoded in mPFC and information about the client's choice encoded in TPJ. I believe they address an important issue and their findings potentially provide significant insights into social neuroscience and other research fields related to human social behavior.

My primary concern is about computer agents' decision-making algorithms (described in page 14). The algorithms look very arbitrary for me. As the authors might know, in a strategic environment participants' behavior would highly depend on the opponents' behavioral algorithm. The authors should therefore justify the artificial algorithms carefully. I would suggest the authors to conduct an additional behavioral experiment with a real human rival advisor and a real human client, and then demonstrate that participants' behaviors in the additional experiment are not significantly different from those in the original experiments.

In Introduction, I am confused by the distinction of "more confident advice", "over-confidence" and "competitive strategy". Do these phrases indicate the same concept?

"However relatively little is known about strategic tracking of social interactions with more than one party". In Introduction and Discussion, it would be worth mentioning previous neuroscience studies on multiplayer ($N > 2$) strategic interactions (e.g., van den Bos et al., JNS, 2013; Suzuki et al., Neuron, 2015).

Page 5. In the first paragraph, the authors could provide a brief summary of the experimental task easier to understand (in other words, the current description is not clear enough).

Page 5. In the first paragraph, the authors show the behavioral data from the three experiments together. I am wondering if behavior in the MRI scanner was consistent with that in the behavioral experiments.

Page 6. The computational model proposed by the authors looks arbitrary. As far as I understand, the model relies on many (implicit) assumptions about participants' belief of the client's decision-making algorithm. I would like the authors to provide further rationale and explanation about the model.

Page 9. While the authors relate TPJ activity to selection by client, the activity can be attributed into the participants' reward signal (given that the participants were rewarded when selected by the client). Is it true? If so, it would be good to discuss the caveat.

Page 10. "Finally, replicating previous works on vicarious reward, we found that the client's reward prediction error at the outcome stage strongly modulated the BOLD responses in left and right Striatum". Although one study (Mobbs et al., Science, 2009) correlated ventral striatum with vicarious reward, according to the meta-analysis (Morelli et al., Neuroimage, 2015) the ventral striatum is significantly more associated with personal reward than vicarious reward. I believe the authors need to revise the current story.

Page 10. "supporting the notion that it has a general role in valuation of various motivational factors that drive social behavior (30)". As far as I know, the ref (30) does not suggest anything about social behavior.

Page 14. For the fMRI experiment, why did the authors recruit male participants only?

Reviewer #2 (Remarks to the Author):

Unlike prior neuroimaging studies of social influence, the authors focus on the agent exerting the influence. This is an interesting perspective that, in my view, significantly extends prior work and could be influential.

The authors developed a novel task that allowed them to evaluate two theories -- competitive strategy and social rank theory -- for acquiring and maintaining social influence. The behavioral results suggest that advisers manage social influence by modulating their advice confidence according to the interaction between their current level of influence (chosen or ignored by the client) and their current relative merit. These behaviors are tracked by distinct brain systems, with MPFC responding to relative merit and TPJ responding to client selection. In addition, both processes involve the ventral striatum.

Although I think the overall question is important and timely, I have several reservations about the results and the extent to which the conclusions are currently supported by the data. These concerns are addressable and are detailed below.

Major Concerns:

1) The task, though elegant and clever, seems to be lacking on ecological validity. Since the authors argue that the adviser's behavior is based on the client and updated dynamically, it would be important to show that an actual human client behaves in the way that the authors programmed the client choice algorithm. The artificial nature of the computerized client's choices raises several questions. For example, is it realistic that the client will always choose the adviser with the higher influence? Do the results generalize to real social interactions? Are the results dependent on social context and social closeness? Addressing these issues would strengthen the results and make the manuscript more broadly appealing.

2) Also related to the task, the noise procedure in Figure S1 isn't clear and I worry that it is artificially driving the observed effects. Is this procedure intended to reduce the

accuracy/confidence of the advice of the affected advisor? In Figure 2C, it seems clear that the noise periods signal when the adviser should adjust their confidence. If the noise periods are omitted, does a similar pattern of behavior emerge?

3) I also worry that the long TR (3.74s) and the timing of the task may not lend itself well to separating different phases of the trial. Would it be possible to show time courses of responses? I think this would be particularly important for Figure 5 where the ventral striatum ostensibly responds to both phases.

Minor Issues:

1) Please remove the non-significant coordinates from the supplemental results. If the coordinates do not survive correction, then they shouldn't be reported in my opinion. Instead, I suggest uploading the thresholded and unthresholded maps to NeuroVault (Gorgolewski et al., 2015, *Frontiers*).

2) A recent paper on social influence isn't cited or discussed, and I believe it could raise an alternative value-based interpretation of the results. Please see Chung et al., 2015, *Nat Neuro* and its associated commentary.

3) The authors hint at the possibility that the results may arise because of an interaction of the brain's social and valuation systems. If this is the case, I wonder why the authors did not consider connectivity analyses? Such analyses have been fruitful in a number of recent papers examining social valuation (e.g., Janowski et al., 2013, *SCAN*; Smith et al., 2014, *SCAN*; van den Bos et al., 2013, *JNeurosci*).

4) If I understand correctly, the trial-to-trial fluctuations in relative merit have magnitude and sign. However, the authors only examine the sign of relative merit (positive vs. negative) and ignore the magnitude. How does magnitude impact the results?

5) In Figure 5, please remove the "independent" ROIs from neurosynth. They are not independent if they were chosen based on the whole-brain results, which seems to be the case right now. In addition, are the effects lateralized or is this a thresholding issue?

6) Equation 5 isn't clear and may contain a typo. Please check all equations for consistency. In addition, there are a number of other typos and mistakes throughout the manuscript, figures, and figure captions. Please review the text carefully before resubmitting.

Reviewer #3 (Remarks to the Author):

This study examines how advice-giving is affected by advice accuracy and social standing in a social decision-making context. In particular, the authors are interested in uncovering when people express overconfidence and when people express lower confidence under two competing hypotheses: "competitive strategy" vs. "defensive strategy". The researchers

manipulated participants' social standing with an advisee who either sought the advice of the participant or a rival advisor, then measured participants' confidence in their own advice. The authors find an interaction between participants' advice accuracy and social standing such that high-accuracy participants who have low social standing have significantly more confidence than participants in any other category. Neural regions, including activation of the mPFC and deactivation of the rTPJ are presented as evidence of processing 'relative merit' and 'client selection'.

Major issues:

Generally, there are a number of aspects of this paper that trouble me. To start with, what are the theoretical motivations of this work? There seems to be a shift in focus from the introduction to the discussion of this manuscript. The introduction is primarily motivated by competing hypotheses about how confidence in advice-giving is influenced by social standing, with hardly any neural predictions made in the introduction. In contrast, the discussion is approached as if the primary research question addressed by this paper is the neural interplay between mPFC and rTPJ, which are suggested to track behavioral measures of advice accuracy and advisee selection, respectively. The manuscript does not make it clear how these research questions are related to each other, or indeed, whether they are related in any substantial way.

By way of example, the discussion section claims that participants follow the "competitive" strategy when they are performing more accurately than their rival advisor, and follow the "defensive" strategy when they are performing less accurately than their rival advisor. This misleadingly suggests a main effect of advice accuracy on confidence ratings, but this is not supported by the interaction data (Fig 3B), unless I am missing something? Indeed, the only main effect demonstrated in this study is not discussed (Fig 2B). This is problematic because it is precisely this interaction between advice accuracy and social standing that motivates the search for neural correlates. It is difficult for me to interpret the finding that rTPJ deactivates during the "evidence" stage. The speculation offered in the manuscript engages in reverse inference and I caution the authors to be more careful here. Furthermore, even the reverse inferences fail to satisfactorily offer plausible psychological mechanisms to explain the neural finding.

Moreover, there is a number of previous studies that readily come to mind that already explore variants of advice giving. Some of this work is cited in the paper, but are embedded in the discussion as mere footnotes. Really this past work should act as the foundation for the current work to build from. For example, both Behrens and Mobbs papers from 2008, 2015, respectively – have probed variants of advice giving at both the behavioural and neural level. Good scholarship should acknowledge this work in the introduction, and discuss how the current work extends this past work.

Like many aspects of this paper, the computational modeling part lacks much needed clarity, particularly in the 'Model Fitting Procedure' section. The raw data (beta/alphas) and fits (BICS) from each of the 5(?) models should be presented. Relatedly, what does the modeling add to this manuscript? To put it another way, what does it help uncover from a behavioral or neural perspective that we did not already know? I ask this, not as a skeptic

of modeling per se, but as someone who can't seem to figure out how it fits into the paper given the current framing. The authors could readily fix this with some re-writing/framing. To briefly summarize my overall thoughts, after reading the manuscript twice, I am still not sure I fully grasp what the major take home message is.

Minor issues:

1. The grammatical errors and awkward sentence constructions throughout this manuscript severely detract from the manuscript's readability. This is more than an aesthetic critique. These grammatical errors and awkward constructions are distracting and often obscure the intended meaning of sentences.
2. Labeling each stage of Figure 1A with their assigned names (e.g. appraisal, evidence, etc.) would greatly help readers who need to reference names/stages in this manuscript.
3. The methods were hard to read and follow, and I found myself re-reading multiple times to gain clarity. For example, was the Fear of Negative Evaluation scale sent a few weeks or a few months after the procedure? Both durations of time are listed in the manuscript. Was this a planned part of the study, or was it performed based on post-hoc hypotheses? How did the subjects know about the contents of the urn and how much confidence they should have (i.e. how were they given this critical information)? These details are not explained, but are crucial for understanding the findings. Moreover, did the subjects see the other's advice after (or before) they made their own decision?
4. The lack of clarity in the methods and theoretical framing (introduction), along with the number of grammar issues, makes this manuscript very difficult to follow and understand. I hope the authors will spend some time re-writing, framing this paper so that it is more digestible to their readers.
5. There are a number of other interesting imaging contrasts that I think would be nice to explore from a computational perspective, including, but not limited to, understanding how learning rates are neurally instantiated within each of the various models.
6. Sample size for an imaging study is very small. I would highly recommend the authors collect a few more subjects so that the $N=20+$. Also 19 were collected but only 14 are reported. It is not clear in the methods why.
7. Although the discussion/warning about social contextual factors (e.g. framing and phrasing) makes sense to me after reading it a few times, I was initially confused because I did not understand its relevance in your discussion. If the authors feel as if the cautionary tale is important, they must better integrate it in the discussion.

Reviewers' comments:

Reviewer #1 (Remarks to the Author):

Hertz et al. have examined how advisors strategically modulate their advice in order to influence their client. The authors have identified key behavioral factors that guide the advisors' decision-making and revealed the neural correlates of the factors: relative merit of their advice (compared with a rival advisor's) encoded in mPFC and information about the client's choice encoded in TPJ. I believe they address an important issue and their findings potentially provide significant insights into social neuroscience and other research fields related to human social behavior.

My primary concern is about computer agents' decision-making algorithms (described in page 14). The algorithms look very arbitrary for me. As the authors might know, in a strategic environment participants' behavior would highly depend on the opponents' behavioral algorithm. The authors should therefore justify the artificial algorithms carefully. I would suggest the authors to conduct an additional behavioral experiment with a real human rival advisor and a real human client, and then demonstrate that participants' behaviors in the additional experiment are not significantly different from those in the original experiments.

Following the reviewer's suggestion, we ran an additional experiment with real human rival advisers and client. All aspects of the experiment were identical to the original one but with live interaction between participants.

Participants arrived in the lab in groups of three, and were told that they were about to play an adviser game together, and that the roles of two advisers and client would be assigned randomly at the beginning of the experiment. Just as in the original experiments, the participants were seated in isolated individual cubicles. They were then randomly assigned the roles of advisers or client for the entire duration of the experiment. As before, they received fixed monetary reward. They played the game for 130 trials.

We collected data from 19 triplets (57 participants, 24 males and 33 females), and had to exclude 3 triplets from analysis. These included 2 triplets in which advisers used only the highest confidence levels (4 and 5), and one triplet in which the client chose only one adviser throughout the entire experiment. We analysed and the data from 32 advisers. We applied the original analysis (including model fitting) to interpret the behaviour as a function of the 2 (Selection by Client) x 2 (Relative Merit) factorial design. The results from the live interaction experiment replicated the original experiment with virtual agents: advice deviance was highest when the adviser felt he was unjustly ignored, i.e. his /her relative merit was positive but the client ignored him/her (P. Fig 3 and below). Using a mixed effects ANOVA we found a significant interaction effect ($F(1,96) = 5.05, p = 0.03$), and a significant effect of relative merit ($F(1,96) = 4.43, p = 0.04$).

Revised Figure 3, including the results from the live interaction experiment.

To further examine the relation between the virtual and live agents experiments, we fitted data from both experiments with another mixed effects ANOVA, with Relative Merit (positive/negative), Selection by Client (chosen/ignored) and Agents (live/virtual) as main factors, and subjects as random effect factor nested within the Agents factor. We did not observe any effect of Agents on advice, neither main effects nor interaction. Overall we observed a significant Relative Merit main effect ($F(1,412) = 8.5$, $p = 0.004$), a significant Selection by Client main effect ($F(1,412) = 5.17$, $p = 0.024$) and a significant Merit*Selection by Client interaction effect ($F(1,412) = 7.24$, $p = 0.008$).

The additional data from live interaction, suggested by the reviewer, support and strengthen the strategic advice giving behaviour we originally observed, showing that this behaviour is not confounded by our algorithms. It also provides an independent replication of our results, demonstrating their robustness (Figure 3C, pages 10-11,23).

In Introduction, I am confused by the distinction of "more confident advice", "over-confidence" and "competitive strategy". Do these phrases indicate the same concept?

We apologize for this confusion. The phrase "overconfidence" refers to advice confidence greater than what a perfectly calibrated adviser would have reported given the information in the black and white grid. "More confident advice" compares the confidence of the rival advisors in a given trial. When using the "competitive strategy", an advisor reports higher confidence (i.e. radical opinion) when s/he is ignored but offers low confidence (i.e. nuanced conservative opinion) when s/he is selected by the client. We have now modified the introduction to clear the confusion and distinguish between confidence, overconfidence and competitive strategy (page 3).

"However relatively little is known about strategic tracking of social interactions with more than one party". In Introduction and Discussion, it would be worth mentioning previous neuroscience studies on multiplayer ($N > 2$) strategic interactions (e.g., van den Bos et al., JNS, 2013; Suzuki et al., Neuron, 2015).

We thank the reviewer for this suggestion. We adapted the introduction, and now included studies investigating the neural mechanisms underlying tracking of multiplayer interactions and social influence (p. 15-16). Interestingly, the suggested papers and others (for example Klucharev et al. 2009 (Klucharev et al., 2009)) are concerned with relationships with a group, and not with multiple relationships between different

individuals. In our case the participant must track two distinctive, inter-dependent relationships, with a client and with the rival adviser, each affecting behaviour in a different manner.

Page 5. In the first paragraph, the authors could provide a brief summary of the experimental task easier to understand (in other words, the current description is not clear enough).

We revised the description of the experimental task, adding more details and procedure information, as well as a link to an online demo (page 5).

Page 5. In the first paragraph, the authors show the behavioral data from the three experiments together. I am wondering if behavior in the MRI scanner was consistent with that in the behavioral experiments.

We had previously provided this data in the supplementary figure S5 (also updated version below, and in supplementary materials, Figure S5). We now added a link to this figure in the main manuscript (page 9). The results from the three cohorts (online, lab and scanner) are presented separately. This analysis showed that the main result holds in all cohorts: advisers gave more radical advice when their relative merit was positive but were ignored by the client (yellow triangle, see the figure below) compared to when the client chose them and they had positive relative merit (* $p < 0.05$, ** $p < 0.005$, **** $p < 0.00005$). However, in respect to negative relative merit, behaviour seemed to be less consistent across cohorts. To formally and statistically test whether there were significant differences between the cohorts, we carried a mixed effects ANOVA (2x2x4) to examine the cohort dependent variations with selection, relative merit and Cohort as main fixed effects, and participants as random effect nested in Cohort. This resulted in significant main effect for relative merit ($F(1, 412) = 15.5, p = 0.0002$), but not main selection effect ($F(1, 412) = 3.52, p = 0.06$) or Cohort effect ($F(2, 412) = 1.12, p = 0.34$). As expected from the repeated main result, interaction between relative merit and selection was significant ($F(1, 412) = 15.75, p = 0.0001$). Interaction between relative merit and Cohort was significant ($F(2, 412) = 5.4, p = 0.001$), but not interaction between selection and Cohort ($F(2, 412) = 0.88, p = 0.45$) and not the triple interaction between relative merit, selection and Cohort ($F(2, 412) = 0.21, p = 0.88$). These suggest that there were some differences between cohorts, but not in the main results, i.e. the interaction effect, thus allowing pooling all the experiments together.

Page 6. The computational model proposed by the authors looks arbitrary. As far as I understand, the model relies on many (implicit) assumptions about participants' belief of the client's decision-making algorithm. I would like the authors to provide further rationale and explanation about the model.

We have now modified the introduction of the trial by trial modelling approach to more explicitly justify the model assumptions (page 7). Importantly, our model is agnostic about participants' beliefs about the selection rule used by the client. We only use the selection by client variable as information available to the participant on each trial (ignored/selected). Our main motivation to use trial by trial modelling approach was to examine the effect of social comparative performance (relative merit) on advice, as social comparison is at the heart of social rank theory (Festinger, 1954). We hypothesised that the ongoing relative merit measure will affect participants' advice confidence, in addition to the effect of being selected or ignored by the client in the current trial, which was previously examined (figure 2). To test this hypothesis, we compared two models one that only takes into account selection by client with our proposed model also included relative merit against the empirical data.

The model comparison showed that the model employing Selection by Client, relative merit and their interaction explained behaviour most successfully even after accounting for different number of parameters. These findings supported the idea that relative merit affects advising behaviour. Our further analyses, namely the 2x2 ANOVA with Relative Merit, Selection by Client and interaction effects (Figure 3, pages 9-10) were aimed at unpacking this contribution.

Page 9. While the authors relate TPJ activity to selection by client, the activity can be attributed into the participants' reward signal (given that the participants were rewarded when selected by the client). Is it true? If so, it would be good to discuss the caveat.

The reviewer points to a possible alternative interpretation of the findings. In our experiment, Selection by Client did not imply any monetary reward (scanner participants received a fixed monetary reward). However, we agree with the reviewer that selection by client is still a positive outcome and could have been subjectively rewarding. However, activity in TPJ was observed in the opposite contrast (ignored > selected) which was motivated by our socially minded hypothesis. Consequently, we believe that the rTPJ activity is more likely to be a correlate of social influence consistent with the surprising information embedded in being ignored. Indeed, we had also reported the ventral striatal (VS) response in the selected>ignored contrast which is consistent with the rewarding implications of being selected. We do agree with the reviewer that the correlational nature of fMRI data does not allow us to exclude the possibility that the rTPJ effect is a response to negative reward experience by being ignored by the client. We have now added this caveat to the manuscript (page 12).

Page 10. "Finally, replicating previous works on vicarious reward, we found that the client's reward prediction error at the outcome stage strongly modulated the BOLD responses in left and right Striatum". Although one study (Mobbs et al., Science, 2009) correlated ventral striatum with vicarious reward, according to the meta-analysis (Morelli et al., Neuroimage, 2015) the ventral striatum is significantly more associated with personal reward than vicarious reward. I believe the authors need to revise the current story.

We are grateful to the reviewer for pointing us to this meta-analysis. We observed ventral striatum activity tracking the outcome (coin) prediction error. In our task this coin is not associated with the participant's reward, but with the client's reward. However, our design is not ideally designed to test vicarious reward. For example we do not contrast rewards to others and rewards to oneself. We therefore cannot provide a strong support for the vicarious reward hypothesis. As the outcome (coin location) provided information about the accuracy of the participants' advice, coin prediction error can be associated with surprise regarding the advice accuracy, i.e. positive surprise when the adviser was cautious and correct, and smaller surprise when the adviser was very confident and correct. We have now revised our interpretation of this result to reflect these considerations and included a reference to the meta-analysis (pages 13-14).

Page 10. "supporting the notion that it has a general role in valuation of various motivational factors that drive social behavior (30)". As far as I know, the ref (30) does not suggest anything about social behavior.

We thank the reviewer for spotting this. This reference (Lebreton et al., 2009) provides evidence about domain general valuation responses in VS, and we meant to cite in addition this review of the role of VS in social behaviour (Báez-Mendoza and Schultz, 2013) to support the social aspects of VS activity presented here. We corrected the citation accordingly.

Page 14. For the fMRI experiment, why did the authors recruit male participants only?

As fMRI data are more costly than lab based and online experiments we wanted to optimize the homogeneity of the participants, and avoid a possibility of task-irrelevant sex-stereotypical behaviours effecting our data (McPherson et al., 2001; Buchan et al., 2008). While our behavioural results did not point to any significant gender effect, following from earlier studies (Ligneul et al., 2016), we opted for a more cautious approach in the neuroimaging data collection. This leaves the possibility that the neural computations underlying advice giving reported here are different for females, a possibility which can be explored. We have added this clarification to the text (page 20).

Reviewer #2 (Remarks to the Author):

Unlike prior neuroimaging studies of social influence, the authors focus on the agent exerting the influence. This is an interesting perspective that, in my view, significantly extends prior work and could be influential.

The authors developed a novel task that allowed them to evaluate two theories -- competitive strategy and social rank theory -- for acquiring and maintaining social influence. The behavioral results suggest that advisers manage social influence by modulating their advice confidence according to the interaction between their current level of influence (chosen or ignored by the client) and their current relative merit. These behaviors are tracked by distinct brain systems, with MPFC responding to relative merit and TPJ responding to client selection. In addition, both processes involve the ventral striatum.

Although I think the overall question is important and timely, I have several reservations about the results and the extent to which the conclusions are currently supported by the data. These concerns are addressable and are detailed below.

Major Concerns:

1) The task, though elegant and clever, seems to be lacking on ecological validity. Since the authors argue that the adviser's behavior is based on the client and updated dynamically, it would be important to show that an actual human client behaves in the way that the authors programmed the client choice algorithm. The artificial nature of the computerized client's choices raises several questions. For example, is it realistic that the client will always choose the adviser with the higher influence? Do the results generalize to real social interactions? Are the results dependent on social context and social closeness? Addressing these issues would strengthen the results and make the manuscript more broadly appealing.

As detailed in the response to the first reviewer's concern and in the manuscript (Figure 3C, pages 10-11,23), we conducted the experiment suggested by the reviewer. We collected data from 19 triplets (57 participants, 24 males and 33 females), and had to exclude 3 triplets from analysis. We analysed the data from 32 advisers and applied the exact same analysis as previously adopted for our virtual experiment. The results from the live interaction experiment replicated the pattern we obtained with the virtual agents: advice deviance was highest when the adviser felt he was unjustly ignored, i.e. when her/his relative merit was positive but the client ignored her/him (P. Fig 3 and below). Using a mixed effects ANOVA we found a significant interaction effect ($F(1,96) = 5, p = 0.03$), and a significant effect of relative merit ($F(1,96) = 4.43, p = 0.04$).

Revised Figure 3, including the results from the live interaction experiment.

To further examine the relation between the virtual and live agents experiments, we fitted data from both experiments with another mixed effects ANOVA, with Relative Merit (positive/negative), Selection by Client (chosen/ignored) and Agents (live/virtual) as main factors, and subjects as random effect factor nested within the Agents factor. We did not observe any effect of Agents on advice, not main effect and not interaction effect. Overall we observed a significant Relative Merit main effect ($F(1,412) = 8.5$, $p = 0.004$), a significant Selection by Client main effect ($F(1,412) = 5.17$, $p = 0.024$) and a significant Merit*Selection by Client interaction effect ($F(1,412) = 7.24$, $p = 0.008$). The results from this new experiment show that it is safe to generalize our experimental model of social influence to more realistic and indeed less controllable case of three interacting participants.

The reviewer is also concerned about the plausibility of real clients behaving according to the algorithm used in the previously submitted manuscript. Running the experiment permitted us to address the question whether the client chooses the adviser with the higher influence. We simulated the choices that our algorithm would have made facing the advice history of the two advisers and the outcome, and compared these simulations with the real choices made by the participants who played the client. An obvious caveat here is that $N=16$ clients was perhaps too small a sample size to test this hypothesis. Notwithstanding the sample size caveat, we found some correspondence between the algorithm and clients' choices (see figure below), as there was a marginally significant correlation between the number of trials that human and virtual clients chose the same adviser ($R = 0.48$, $p = 0.084$). This finding indicates that the algorithm may bare similarity to real clients, capturing some basic elements of the human clients' behaviour. Both real clients and the algorithm's choices of advisers were influenced by the adviser's history of accuracy and confidence. Needless to say, the study of the client's behaviour falls under the category of advice taking and opinion aggregation (see manuscript introduction). We therefore believe that this interesting and important question, which merits its own structured investigation with an experimental design tailored to examine different strategies of client behaviour falls outside the focus of the current work.

Finally, the reviewer asks how our results depend on social context and social closeness. A recent study showed that persuasion between friends depends on the direction of friendship, e.g. mutual or unidirectional from adviser to advisee (Almaatouq et al., 2016). It is also suggested that gender may affect the use of confidence in advise giving, as females are expected to display lower confidence than males (Bowles et al., 2007), and therefore may opt to use different persuasion strategies. In our experiment, we recruited participants separately and debriefings confirmed that they did not know each other. Consequently, our measurements and results do not offer a strong hypothesis for the role of closeness. However, our laboratory model and current results provide a benchmark against which future investigation of this question can be compared. For example, one may manipulate advisers/clients gender in order to examine gender effect, or elicit in-group out-group dynamics by providing some background information about the client and advisers. We included this suggestion in the discussion section of the manuscript (page 18).

2) Also related to the task, the noise procedure in Figure S1 isn't clear and I worry that it is artificially driving the observed effects. Is this procedure intended to reduce the accuracy/confidence of the advice of the affected advisor? In Figure 2C, it seems clear that the noise periods signal when the adviser should adjust their confidence. If the noise periods are omitted, does a similar pattern of behavior emerge?

We revised the description of the noise procedure, and extended figure S1 (Figure S1, pages 23-24). In our early pilots we noticed that sometimes, the virtual client did not shift between participants, practically ignoring one of them throughout the experiment. This happened because advisers tended to be similarly calibrated with the evidence. As we intended to use a within-participants design to compare advice on periods in which the adviser is chosen and periods in which he is ignored by the client, we needed some manipulation to increase the probability of differences in advice and accuracy between advisers, which would then entail client switching between advisers. We therefore introduced noise to one of the advisers' evidence, i.e. the ratio between black and white squares in the grid (see Figure 1). The noise procedure went as follows. If the probability of the coin being in the black urn on a specific trial was 0.75, the grid would normally include 75 black squares and 25 white squares. On a noisy trial, this composition changed to 55 black squares and 45 white squares, i.e. reduction of contrast by 20 squares. Similarly, when the probability of the coin being in the *white* urn on a specific trial was 0.75 (0.25 probability of being in the black), the noisy grid would include 55 white squares instead of 75 white squares in the not noisy case, again reducing the contrast by 20 squares. In all noisy trials' contrasts were reduced by 20 squares in a similar fashion. Importantly, noisy periods were relatively short, lasting 10 consecutive trials in the online experiment and 5 consecutive trials in the lab and scanner experiments. Noise was introduced either to the participant or to the other adviser (i.e. the virtual rival algorithm was fed noisy evidence). Online experiments included 4 blocks of noisy periods, 2 for each adviser, while the longer lab based and scanner experiments included 8 blocks of noisy period, 4 for each adviser.

Finally, we also followed the reviewer's suggestion and re-analysed the data after omitting the noise periods. Our results did not change (compare the panels below), with a significant main effect of Relative Merit ($F(1,315) = 5.29, p = 0.02$), and significant Relative Merit x Selection by Client effect ($F(1,315) = 13.1, p = 0.0005$) (See figure below). We now added this figure and results to supplementary materials (Figure S1).

3) I also worry that the long TR (3.74s) and the timing of the task may not lend itself well to separating different phases of the trial. Would it be possible to show time courses of responses? I think this would be particularly important for Figure 5 where the ventral striatum ostensibly responds to both phases.

We agree that the long TR used in this study, chosen for acquiring whole-brain coverage, is less than optimal for detecting brain activity locked to different stages (or phases) within the same trial. To compensate for the long TR we used a large number of events (240 trials x 6 stages = 1440 events, with jitter in timing between stages) in a fast event related design, which supported the need for dynamic and engaging task and allow some degree of stages or phases separation within a trial. We would caution the reader that the time course of brain activity measured by our procedure might be too coarse to lock brain activity to specific stages within a trial. However, we maintain that the temporal order of the observed activities should be detectable and consistent, as in figure 5 where activity in the striatum started with coding relative merit first and then Selection by Client.

As requested, we have plotted the time courses from the ventral striatum activity to demonstrate the temporal order of activity in the right and left Ventral Striatum. We followed the steps used in previous studies (for example in Behrens et al. 2008 (Behrens et al., 2008)) to examine time courses of trials containing multiple jittered stages. We separated each subject's time series sampled from the VS into each trial, and resampled each trial to a duration of 15s: previous trial other's advice (stage 5) at time 0, previous trial outcome (stage 6) at time 2s, Selection by Client (stage 1) at time 4s, evidence (stage 2) at time 6s, confidence report (stages 3-4) at time 9s-11s, and current trial other's advice (stage 5) at time 13s (These timings were the mean timings across all trials in all subjects.) The resampling resolution was 100ms. This temporal realignment allowed the observation of signal throughout the trial while taking advantage of the random jitter and fast event-related design. Importantly, this procedure captures the temporal order of BOLD signal. Following the convention (Behrens et al., 2008), we performed a GLM in each time point across trials in each subject. We had one regressor for Selection by Client (chosen/ignored), and another one for relative merit sign (positive/negative). We then calculated the mean of the effect across subjects at each time point, and their standard errors. This GLM approach does not depend on separation of trials for the four events mentioned above, as each trial is decoded according to its relative merit and Selection by Client properties.

The plotted regression results replicate the temporal order reported in the manuscript, an order that was captured by our standard whole brain GLM analysis. A positive effect of relative merit (pink lines) is observed first followed by a later effect of Selection by Client (green line).

We added this figure and methods to the main text (Figure 5, page 27).

Minor Issues:

1) Please remove the non-significant coordinates from the supplemental results. If the coordinates do not survive correction, then they shouldn't be reported in my opinion. Instead, I suggest uploading the thresholded and unthresholded maps to NeuroVault (Gorgolewski et al., 2015, Frontiers).

We removed the non-significant coordinates from the supplemental tables, and uploaded the maps to NeuroVault as suggested: <http://neurovault.org/collections/2204/>

2) A recent paper on social influence isn't cited or discussed, and I believe it could raise an alternative value-based interpretation of the results. Please see Chung et al., 2015, Nat Neuro and its associated commentary.

We thank the reviewer for bringing this paper to our attention. We have now addressed this study and others looking at how one is influenced by other's actions and choices (Lebreton et al., 2012) in the Discussion of the revised manuscript (page 4).

3) The authors hint at the possibility that the results may arise because of an interaction of the brain's social and valuation systems. If this is the case, I wonder why the authors did not consider connectivity analyses? Such analyses have been fruitful in a number of recent papers examining social valuation (e.g., Janowski et al., 2013, SCAN; Smith et al., 2014, SCAN; van den Bos et al., 2013, JNeurosci).

We thank the reviewer for the suggestion and references. Following these references, we examined our functional connectivity hypothesis using PPI (psychophysiological interaction) approach. To reduce the degrees of freedom of our analysis, and to avoid multiple exploratory analyses, we restricted our analysis to the cortical and subcortical areas identified in the whole brain analysis. Furthermore, we examined PPI connectivity only with respect to the stages and contrasts in which whole brain analysis was carried out. We hypothesised that changes of merit at the end of a trial may affect the brain response to client choices in the following trial, an effect that may underlie the behavioural interaction effect. We therefore examined the PPI connectivity between mPFC and rTPJ during outcome stage, and between the left VS and rTPJ

during the Selection by Client stage. We contrasted PPI coefficients between positive merit trials and negative merit trials. We found a significant connectivity between mPFC and rTPJ, which was higher for positive merit trials compared with negative merit trials ($t(17) = 3.25$, $p = 0.004$). In addition, individual PPI coefficients were significantly correlated with the weight of merit on advice deviance (β_{Merit}), estimated by our interaction model ($R(17) = 0.66$, $R^2 = 0.44$, $p = 0.003$). VS connectivity with rTPJ was not significant ($t(17) = 0.7$, $p = 0.5$), and the coefficients were not correlated with the behavioural model estimations. Increased connectivity between mPFC and rTPJ during positive merit trials provides a mechanism for integrating the information about merit and selection by client, which explain the interaction effect we observed in the behavioural results. When merit is positive, rTPJ may be more sensitive to being ignored. The fact that this connectivity is correlated with the estimated model parameters for the merit effect supports that notion. Interestingly, functional connectivity was observed only between cortical areas and not between VS and rTPJ, i.e. within the social brain network. We now added this result to the manuscript (Figure 6, pages 14,27).

4) If I understand correctly, the trial-to-trial fluctuations in relative merit have magnitude and sign. However, the authors only examine the sign of relative merit (positive vs. negative) and ignore the magnitude. How does magnitude impact the results?

To address this question, we fitted an interaction model, which used the sign *and* amplitude of the relative merit to the participants' advice deviance:

$$AdviceDeviance(t) = Bias + \beta_{Selection} \cdot Selection(t) + \dots$$

$$\beta_{Merit} \cdot RelativeMerit(t) + \beta_{Interaction} \cdot Selection(t) \cdot RelativeMerit(t)$$

This model performed worse than the interaction model using only the sign of the relative merit (mean±std DIC sign and amplitude: 83±42, DIC sign only: 80±42, paired ttest: $t(105) = 6.2$, $p = 10^{-9}$, Effect Size = 0.6). The parameters estimated for the weight of Selection by Client and relative merit did not differ between the two models, with the weight of Selection by Client still being significantly lower than zero ($t(105) = 3.8$, $p = 0.0001$). However, the learning rate of relative merit was lower on average for the amplitude model, maybe forcing some normalization on the trial-by-trial changes in relative merit. These results suggest that the amplitude of relative merit may have less effect on advice deviance, and a categorical positive/negative differentiation better explains participants' behaviour. This is now included in supplementary materials (Supplementary Figure S2).

5) In Figure 5, please remove the “independent” ROIs from neurosynth. They are not independent if they were chosen based on the whole-brain results, which seems to be the case right now. In addition, are the effects lateralized or is this a thresholding issue?

Using whole brain analysis, we identified the VS as areas coding Selection by Client and Relative Merit effects. We decided to use a meta-analysis based definition of the VS, to provide an external indication that responses to relative merit and Selection by Client overlapped in location with previously reported responses to monetary rewards and prediction errors. The ROIs (left and right VS) were defined independently from our analysis, as a sphere around the coordinates from Neurosynth, but the motivation to examine activity in these ROIs was based on the whole brain analysis.

We appreciate that this may not qualify as “independent” ROIs, however we feel that there still something to be gained from the use of Neurosynth defined VS ROIs. We removed definition “independent” from the Neurosynth ROIs and gave a detailed description of our motivation for use of Neurosynth ROIs in the main text (Figure 5, pages 13,26).

The effects we observed were apparent bilaterally, but each were more pronounced in one side, while the other side did not survive our cluster size correction for multiple comparisons. In the figure below we added the maps with no cluster size correction (and slightly reduced threshold), showing that activation could also be seen in the other side of VS. We uploaded these maps to NeuroVault, so readers can explore the uncorrected maps (<http://neurovault.org/collections/2204/>). While our results support a temporal difference in responses to merit and Selection by Client in the VS, they provide weak evidence for lateralization. We now made this distinction explicit in the manuscript (page 13).

6) Equation 5 isn't clear and may contain a typo. Please check all equations for consistency. In addition, there are a number of other typos and mistakes throughout the manuscript, figures, and figure captions. Please review the text carefully before resubmitting.

Thank you for pointing the inaccuracies in the equation. We examined equation 5 again and adjusted it and equation 4 for clarity. This equation describes the cost function, $L(M)$ used to estimate a model M fit to the data. The cost function compares the advice deviance estimated with a model M ($AdviceDeviance_M(t)$) and the actual advice deviance made by the participants on each trial ($AdviceDeviance_{Data}(t)$):

$$[5] \quad L(M) = - \sum_{t=1..T} \log \left(\frac{1}{1 + abs(AdviceDeviance_M(t) - AdviceDeviance_{Data}(t))} \right)$$

Like log likelihood cost function, the ratio inside the log will be close to one when the estimation is close to the data, and it gets closer to zero when the distance between estimation and data increases. As such, lower cost function value indicates better fit of the model to the data. This clarification is now indicated in the manuscript (page 24).

In addition, all authors reviewed the manuscript, figures and figure captions for typos and mistakes, and hope there are none left.

Reviewer #3 (Remarks to the Author):

This study examines how advice-giving is affected by advice accuracy and social standing in a social decision-making context. In particular, the authors are interested in uncovering when people express overconfidence and when people express lower confidence under two competing hypotheses: “competitive strategy” vs. “defensive strategy”. The researchers manipulated participants’ social standing with an advisee who either sought the advice of the participant or a rival advisor, then measured participants’ confidence in their own advice. The authors find an interaction between participants’ advice accuracy and social standing such that high-accuracy participants who have low social standing have significantly more confidence than participants in any other category. Neural regions, including activation of the mPFC and deactivation of the rTPJ are presented as evidence of processing ‘relative merit’ and ‘client selection’.

Major issues:

Generally, there are a number of aspects of this paper that trouble me. To start with, what are the theoretical motivations of this work? There seems to be a shift in focus from the introduction to the discussion of this manuscript. The introduction is primarily motivated by competing hypotheses about how confidence in advice-giving is influenced by social standing, with hardly any neural predictions made in the introduction. In contrast, the discussion is approached as if the primary research question addressed by this paper is the neural interplay between mPFC and rTPJ, which are suggested to track behavioral measures of advice accuracy and advisee selection, respectively. The manuscript does not make it clear how these research questions are related to each other, or indeed, whether they are related in any substantial way.

We revised the manuscript to make it more cohesive. In the introduction, we now clarify our behavioural as well as neurobiological hypotheses. We now describe the different computations associated with the two competing hypothesis: the competitive strategy relies mostly on tracking the client’s responses and choices, computations that were identified in the rTPJ and whose valence is tracked in the ventral striatum (Pagnoni et al., 2002; van Schie et al., 2004; Hampton et al., 2008; Behrens et al., 2009; Deen et al., 2015; Mobbs et al., 2015), , while the defensive strategy, and indeed social rank theory literature, highlights the internal process of self-comparisons with peers, computations associated with activity in prefrontal cortex and parietal cortex (Frith and Frith, 2003; Gilbert et al., 2006; Kumaran et al., 2016; Ligneul et al., 2016)(page 3). This description helps provide neural predictions associated with these different processes in the introduction (pages 3-4), and to link the neural results, behavioural results and our hypotheses (page 15).

In the Discussion, we now discuss the behavioural and neurobiological findings together and offer our account of how they relate to each other.

By way of example, the discussion section claims that participants follow the “competitive” strategy when they are performing more accurately than their rival advisor, and follow the “defensive” strategy when they are performing less accurately than their rival advisor. This misleadingly suggests a main effect of advice accuracy on confidence ratings, but this is not supported by the interaction data (Fig 3B), unless I am missing something? Indeed, the only main effect demonstrated in this study is not discussed (Fig 2B). This is problematic because it is precisely this interaction between advice accuracy and social standing that motivates the search for neural correlates.

We revised the summary of the results by noting that participants follow the “competitive” strategy when they are performing more accurately than their rival advisor, but not when they are performing less accurately than their rival advisor. Thanks to the reviewer comment, we agree this is a more accurate description of the interaction effect we observed. We now clarify that the main effect of being selected by the client does not provide an exhausting account of advisers’ behaviour in our task and point out that the interaction effect reveals that Selection by Client (ignored/chosen) affected behaviour only when relative merit was positive. Furthermore, this interaction effect, although not the main effect of Selection by Client, was replicated in the live experiment where the two advisers and the client were played by real people (see reviewer 1 and 2 comments). We have now clarified the interpretation of the behavioural results (page 15).

It is difficult for me to interpret the finding that rTPJ deactivates during the “evidence” stage. The speculation offered in the manuscript engages in reverse inference and I caution the authors to be more careful here. Furthermore, even the reverse inferences fail to satisfactorily offer plausible psychological mechanisms to explain the neural finding.

We apologise for the unfortunate choice of colour scale and sign of comparison in the rTPJ figure (previous Fig. 3B). The observed effect is not a deactivation, i.e. decreased activity below baseline, but increased activity when the participant is ignored by the client, compared to trials in which the participant was the chosen adviser (Ignored > Chosen), as can be seen in the plot of beta values (revised Fig. 3, and below). This effect is in line with the literature, demonstrating increased activity in the rTPJ on trials in which others’ behaviour does not match one’s beliefs about their intentions (Koster-Hale and Saxe, 2013), or when detecting someone lying (Behrens et al., 2008), and when inferring how much we influence another person’s behaviour (Hampton et al., 2008). Our interpretation of this result is that the rTPJ tracks the current influence level the adviser has over the client, which in interaction with Relative Merit (pages 11-12 of revised manuscript) gives rise to the manifestation of the competitive strategy i.e. overstating the perceptual evidence when ignored by the client. We have changed the figure (see below) and modified our interpretation of the results to avoid committing a reverse inference.

Moreover, there is a number of previous studies that readily come to mind that already explore variants of advice giving. Some of this work is cited in the paper, but are embedded in the discussion as mere footnotes. Really this past work should act as the foundation for the current work to build from. For example, both Behrens and Mobbs papers from 2008, 2015, respectively – have probed variants of advice giving at both the behavioural and neural level. Good scholarship should acknowledge this work in the introduction, and discuss how the current work extends this past work.

In the revised introduction we now acknowledge previous work on advice taking (Behrens et al., 2008; Biele et al., 2011; Izuma and Adolphs, 2013) and the only work on advice giving (Mobbs et al., 2015). We now discuss the study by Mobbs et al. (pages 4, 17-18) and note that they used advice giving to study the

participants' reflections about self rather than, as we did here, as means to exert social influence. As noted above, we also discuss other literature describing the neural computations linked to strategic advice giving such as tracking one's status and rank (Kumaran et al., 2016) and inferring others' intentions from their actions (Koster-Hale and Saxe, 2013).

Like many aspects of this paper, the computational modeling part lacks much needed clarity, particularly in the 'Model Fitting Procedure' section. The raw data (beta/alphas) and fits (BICS) from each of the 5(?) models should be presented. Relatedly, what does the modeling add to this manuscript? To put it another way, what does it help uncover from a behavioral or neural perspective that we did not already know? I ask this, not as a skeptic of modeling per se, but as someone who can't seem to figure out how it fits into the paper given the current framing. The authors could readily fix this with some re-writing/framing.

The modelling approach used here helped uncover the effect of relative merit, i.e. comparison with rival adviser, on advice giving. Initially our results were analysed only with regards to selection by client, where a main effect was observed (Figure 2). This can be seen as a simple model, in which advice deviance is only determined by selection by client. We compared this model to other, more complicated ones to see whether more detailed models can give better account of the data while compensating for the number of parameters (pages 7-8, 24-25). This procedure showed that a model including selection by client, relative merit, and specially the interaction between relative merit and Selection by Client accounted for the advice deviance much better. Our modelling approach uncovered the significant role of relative merit – which was not self-evident in the raw data analysis, and allowed us to define the trial-by-trial relative merit, used in further analysis to demonstrate the interaction effect and in the analysis of neuroimaging data. We now stress the contribution of the modelling approach. In addition, we provide the parameters estimated for each model in addition to the fits of each model in the supplementary materials (Supplementary materials, Table ST1).

To briefly summarize my overall thoughts, after reading the manuscript twice, I am still not sure I fully grasp what the major take home message is.

We revised the manuscript thoroughly, as suggested above in regards to linking the behavioural hypotheses to their neural predictions and in addition as suggested here by better framing and better description of our results and conclusions. Importantly we revised the first part of the discussion to provide a better summary of the results (pages 15-16).

Minor issues:

1. The grammatical errors and awkward sentence constructions throughout this manuscript severely detract from the manuscript's readability. This is more than an aesthetic critique. These grammatical errors and awkward constructions are distracting and often obscure the intended meaning of sentences.

All authors reviewed the revised manuscript, figures and figure captions for grammatical errors, typos and mistakes, and hope there are none left. We also revised the manuscript heavily, and hope that it is now more readable and clear.

2. Labeling each stage of Figure 1A with their assigned names (e.g. appraisal, evidence, etc.) would greatly help readers who need to reference names/stages in this manuscript.

This is a great suggestion and we updated the figure accordingly (Figure 1).

3. The methods were hard to read and follow, and I found myself re-reading multiple times to gain clarity. For example, was the Fear of Negative Evaluation scale sent a few weeks or a few months after the procedure? Both durations of time are listed in the manuscript. Was this a planned part of the study, or was it performed based on post-hoc hypotheses? How did the subjects know about

the contents of the urn and how much confidence they should have (i.e. how were they given this critical information)? These details are not explained, but are crucial for understanding the findings. Moreover, did the subjects see the other's advice after (or before) they made their own decision?

We revised the methods section, and the description of the task in the main manuscript.

The Fear of Negative Evaluation scale was sent six months after the main experiment, based on hypothesis that behaviour may be explained by traits associated with submissive behaviour (pages 9, 12, 28).

Participants saw a grid of black and white squares for half a second (stage 2, Figure 1). The ratio between the number of black squares and the total number of squares corresponded to the probability that the coin was in the black urn. The content of the chosen urn was revealed at the end of the trial (stage 6, figure 1), and the urn containing the coin could be inferred. The participants were not instructed about how to give their confidence reports. They were given a description of the confidence scale, and were told that selecting high numbers on the confidence scale (5B/5W) is associated with high certainty about the coin location, and low confidence is associated with low certainty about the coin location.

The subjects saw the other's advice after they made their own decision (stage 5, Figure 1).

To clarify the task procedure we provide a link to the web based experiment that was used for the online cohort of this study (<http://www.urihertz.net/AdviserExperiment.html> Page 5), as well as a more detailed description of the task in the beginning of the results section (page 5). The lab based and scanner based participants followed the same experimental procedure except that instructions were given by the experimenter, and the web based experiment started directly on the first trial. We have now added these details to the manuscript to clarify the methods

4. The lack of clarity in the methods and theoretical framing (introduction), along with the number of grammar issues, makes this manuscript very difficult to follow and understand. I hope the authors will spend some time re-writing, framing this paper so that it is more digestible to their readers.

All authors reviewed the revised manuscript, figures and figure captions for grammatical errors, typos and mistakes, and hope there are none left. We also revised the manuscript thoroughly, and hope that it is now more readable and clear.

5. There are a number of other interesting imaging contrasts that I think would be nice to explore from a computational perspective, including, but not limited to, understanding how learning rates are neurally instantiated within each of the various models.

We share the reviewer's sentiment that this is a complex and fascinating dataset, and that there are other interesting contrasts to explore. As this study uses an original task and provides novel behavioural results on advice giving, we examined the neuroimaging contrasts that follow most directly our behavioural analyses and results. We will continue to examine more refined aspects of the data and advice giving behaviour in the future.

6. Sample size for an imaging study is very small. I would highly recommend the authors collect a few more subjects so that the N=20+. Also 19 were collected but only 14 are reported. It is not clear in the methods why.

We collected data from 19 participants in the scanner, and used 18 of them for all the neuroimaging contrasts (one participant's data was corrupted). We obtain 'Fear of Negative Evaluation' scores from 14 of those participants, and used these scores for the correlation with rTPJ effects in Figure 4D, hence the legend N=14 in this panel. We have now clarified this issue in the manuscript (pages 12,20,28).

7. Although the discussion/warning about social contextual factors (e.g. framing and phrasing) makes sense to me after reading it a few times, I was initially confused because I did not

understand its relevance in your discussion. If the authors feel as if the cautionary tale is important, they must better integrate it in the discussion.

We revised this section of the discussion and provided a better motivation for it, which relates it to the rest of the discussion (pages 17-18)

References

- Almaatouq A, Radaelli L, Pentland A, Shmueli E (2016) Are you your friends' friend? Poor perception of friendship ties limits the ability to promote behavioral change. *PLoS One* 11:1–13.
- Báez-Mendoza R, Schultz W (2013) The role of the striatum in social behavior. *Front Neurosci* 7:1–14 Available at: <http://journal.frontiersin.org/article/10.3389/fnins.2013.00233/abstract>.
- Behrens TEJ, Hunt LT, Rushworth MFS (2009) The computation of social behavior. *Science* 324:1160–1164 Available at: <http://www.ncbi.nlm.nih.gov/pubmed/19478175> [Accessed March 9, 2012].
- Behrens TEJ, Hunt LT, Woolrich MW, Rushworth MFS (2008) Associative learning of social value. *Nature* 456:245–249 Available at: <http://www.pubmedcentral.nih.gov/articlerender.fcgi?artid=2605577&tool=pmcentrez&rendertype=abstract> [Accessed March 1, 2012].
- Biele G, Rieskamp J, Krugel LK, Heekeren HR (2011) The neural basis of following advice. *PLoS Biol* 9:e1001089 Available at: <http://www.pubmedcentral.nih.gov/articlerender.fcgi?artid=3119653&tool=pmcentrez&rendertype=abstract> [Accessed November 8, 2013].
- Bowles HR, Babcock L, Lai L (2007) Social incentives for gender differences in the propensity to initiate negotiations: Sometimes it does hurt to ask. *Organ Behav Hum Decis Process* 103:84–103 Available at: <http://linkinghub.elsevier.com/retrieve/pii/S0749597806000884>.
- Buchan NR, Croson RT a, Solnick S (2008) Trust and gender: An examination of behavior and beliefs in the Investment Game. *J Econ Behav Organ* 68:466–476.
- Deen B, Koldewyn K, Kanwisher N, Saxe R (2015) Functional Organization of Social Perception and Cognition in the Superior Temporal Sulcus. *Cereb Cortex*:1–14 Available at: <http://www.cercor.oxfordjournals.org/cgi/doi/10.1093/cercor/bhv111>.
- Festinger L (1954) A Theory of Social Comparison Processes. *Hum Relations* 7:117–140 Available at: <http://hum.sagepub.com/cgi/doi/10.1177/001872675400700202>.
- Frith U, Frith CD (2003) Development and neurophysiology of mentalizing. *Philos Trans R Soc Lond B Biol Sci* 358:459–473 Available at: <http://www.ncbi.nlm.nih.gov/pubmed/12689373>.
- Gilbert SJ, Spengler S, Simons JS, Steele JD, Lawrie SM, Frith CD, Burgess PW (2006) Functional Specialization within Rostral Prefrontal Cortex (Area 10): A Meta-analysis. *J Cogn Neurosci* 18:932–948 Available at: <http://www.ncbi.nlm.nih.gov/pubmed/16839301>.
- Hampton AN, Bossaerts P, O'Doherty JP (2008) Neural correlates of mentalizing-related computations during strategic interactions in humans. *Proc Natl Acad Sci U S A* 105:6741–6746 Available at: <http://www.pubmedcentral.nih.gov/articlerender.fcgi?artid=2373314&tool=pmcentrez&rendertype=abstract> [Accessed January 27, 2014].
- Izuma K, Adolphs R (2013) Social manipulation of preference in the human brain. *Neuron* 78:563–573 Available at: <http://www.pubmedcentral.nih.gov/articlerender.fcgi?artid=3695714&tool=pmcentrez&rendertype=abstract> [Accessed July 17, 2014].
- Klucharev V, Hytönen K, Rijpkema M, Smidts A, Fernández G (2009) Reinforcement Learning Signal Predicts Social Conformity. *Neuron* 61:140–151 Available at: <http://linkinghub.elsevier.com/retrieve/pii/S0896627308010209>.
- Koster-Hale J, Saxe R (2013) Theory of Mind: A Neural Prediction Problem. *Neuron* 79:836–848 Available at: <http://linkinghub.elsevier.com/retrieve/pii/S089662731300754X>.
- Kumaran D, Banino A, Blundell C, Hassabis D, Dayan P (2016) Computations Underlying Social

Hierarchy Learning: Distinct Neural Mechanisms for Updating and Representing Self-Relevant Information. *Neuron* 92:1135–1147 Available at:
<http://linkinghub.elsevier.com/retrieve/pii/S0896627316308029>.

Lebreton M, Jorge S, Michel V, Thirion B, Pessiglione M (2009) An Automatic Valuation System in the Human Brain: Evidence from Functional Neuroimaging. *Neuron* 64:431–439.

Lebreton M, Kawa S, Forgeot d'Arc B, Daunizeau J, Pessiglione M (2012) Your Goal Is Mine: Unraveling Mimetic Desires in the Human Brain. *J Neurosci* 32:7146–7157.

Ligneul R, Obeso I, Ruff C, Dreher J (2016) Dynamical representation of dominance relationships in the human medial prefrontal cortex. *Curr Biol* 1:1–33 Available at:
<http://dx.doi.org/10.1016/j.cub.2016.09.015>.

McPherson M, Smith-Lovin L, Cook JM (2001) Birds of a Feather: Homophily in Social Networks. *Annu Rev Sociol* 27:415–444 Available at:
<http://www.annualreviews.org/doi/abs/10.1146/annurev.soc.27.1.415>.

Mobbs D, Hagan CC, Yu R, Takahashi H, FeldmanHall O, Calder AJ, Dalgleish T (2015) Reflected glory and failure: the role of the medial prefrontal cortex and ventral striatum in self vs other relevance during advice-giving outcomes. *Soc Cogn Affect Neurosci* 10:1323–1328 Available at:
<http://scan.oxfordjournals.org/lookup/doi/10.1093/scan/nsv020>.

Pagnoni G, Zink CF, Montague PR, Berns GS (2002) Activity in human ventral striatum locked to errors of reward prediction. *Nat Neurosci* 5:97–98 Available at:
<http://www.ncbi.nlm.nih.gov/pubmed/11802175>.

van Schie HT, Mars RB, Coles MGH, Bekkering H (2004) Modulation of activity in medial frontal and motor cortices during error observation. *Nat Neurosci* 7:549–554.

Reviewers' comments:

Reviewer #1 (Remarks to the Author):

The authors have adequately addressed my concerns, especially the inclusion of the new live interaction experiment.

Reviewer #2 (Remarks to the Author):

The authors have done an excellent job with their response and revisions. My original comments have been largely addressed, but I have a few remaining (minor) issues that merit additional consideration:

The additional experiment is helpful and it is good to see that there is a marginally significant correlation between the number of trials that human and virtual clients chose the same adviser. However, is not the case that the means differ? Judging from the scatterplot, it looks like mean of the x-axis (real client) is much higher than the mean of the y-axis (algorithm client).

I agree with Reviewer 3 that the sample size is very small, and I regret not flagging this in my original review. It is unfortunate that the authors did not respond to this point, which is becoming increasingly salient in the neuroimaging community (Poldrack et al., 2017, *Nat Rev Neuro*). And frankly, it is a bit peculiar that the authors say that $N=16$ is "perhaps too small" and then settle for $N=18$ for their key imaging results. The authors could have explained the sample size choice by pointing to a large effect size in an independent pilot neuroimaging study. In addition, the authors could have also explained why they didn't collect additional data since the time of the first review. At a minimum, it would be helpful to see 95% confidence intervals plotted around each effect so that the reader has a better impression about the uncertainty associated with the effect (e.g., Ingre, 2013, *Neuroimage*; Lindquist et al., 2013, *Neuroimage*).

Please avoid any points about lateralization unless it is explicitly tested. It doesn't seem to be an important issue at all for this study, so I suggest completely avoiding it.

The addition of the PPI analysis helps illustrate how these regions are interacting in various parts of the task. However, the description of the PPI model wasn't completely clear to me. It seems like the authors only had one PPI regressor, which wouldn't be adequate to capture the conditions in this design. Please consider showing that the results hold with the generalized PPI model (McLaren et al., 2012, *Neuroimage*). Also note that similar interactions between MPFC and TPJ were reported in a recent meta-analysis of PPI studies (Smith et al., 2016, *HBM*), which might lend greater confidence to the authors observed effects.

I'm glad the authors clarified the procedures and rationale underlying the noise periods in the task. However, it is not clear to me how "omitting the noise" from the analyses

addresses my original point about the noise driving behavior. Please explain further.

Reviewer #3 (Remarks to the Author):

I applaud that the authors ran a subsequent study that includes all real participants, as I think this strengthens the manuscript significantly, however I still have a number of concerns.

First, I could not find many of the revisions in the manuscript that the authors said they did. For example, the authors say that they substantially changed the introduction to better address the theoretical motivations for the mpfc and tpj. This was not done. The *only* addition is on page 2 where they write "...evaluation of other's performance and comparing it with that of oneself, a computation underpinned by the mPFC." But *why* cite the mPFC here? Simply referencing this brain region does not mean that it has been properly motivated. In line with this, the authors also state that they now integrated the previous work I referenced in my first round of reviews as relevant to their topic. Again, I cannot find where this was done.

Second, it is still not clear to me why the authors are arguing that in a modelling paper they do not want to fit their learning rates to the MRI data? In line with this, their ST1 table does not include alphas and it should.

Third, the manuscript is still rife with grammar issues. There is about one on every page. For example, on page 2 alone, there is a sentence that reads "...the theory suggests people may response according to their rank in a group." Please read over your manuscript carefully before submitting as these mistakes are easy to fix and only serve to annoy the readers.

Fourth, I do think we should be staying away from calling a network of brain regions the 'social brain system' (pg 2). That 'social brain system' is the same system that encodes many non-social phenomena as well. Again, this type of language boils down to proper motivation of understanding why certain brain regions may be involved in processing this type of advice giving.

Fifth, how was the PPI done? What was the seed region? Again, more details about these analyses must be reported.

Reviewers' comments

Reviewer #1 (Remarks to the Author)

The authors have adequately addressed my concerns, especially the inclusion of the new live interaction experiment.

Reviewer #2 (Remarks to the Author)

The authors have done an excellent job with their response and revisions. My original comments have been largely addressed, but I have a few remaining (minor) issues that merit additional consideration:

The additional experiment is helpful and it is good to see that there is a marginally significant correlation between the number of trials that human and virtual clients chose the same adviser. However, is not the case that the means differ? Judging from the scatterplot, it looks like mean of the x-axis (real client) is much higher than the mean of the y-axis (algorithm client).

We directly examined the difference in choosing adviser 1 between the real clients and the algorithm controlled clients, and did not find a significant difference: Mean_Algorithm = 56.28, Mean_Real_Client = 68.85, $t(15) = 1.45$, $p = 0.17$. This does not mean that there are no discrepancies between our algorithm behaviour and human client behaviour, but it suggests that our algorithm does not differ very much from live clients.

We added this information in the revised version of the manuscript (page 11 and Figure S6 in Supplementary Materials).

Importantly, in the real clients experiment, advisers' behaviour remained very similar to the one displayed in the virtual clients experiment, showing that strategic changes in advice confidence were not a mere adaptation to the specific manner in which our algorithm was coded. In general we agree that the computational dynamics governing the client's behaviour are very interesting, and accordingly we intend to explore them in more rigorous manner in dedicated follow-up studies.

I agree with Reviewer 3 that the sample size is very small, and I regret not flagging this in my original review. It is unfortunate that the authors did not respond to this point, which is becoming increasingly salient in the neuroimaging community (Poldrack et al., 2017, Nat Rev Neuro). And frankly, it is a bit peculiar that the authors say that N=16 is "perhaps too small" and then settle for N=18 for their key imaging results. The authors could have explained the sample size choice by pointing to a large effect size in an independent pilot neuroimaging study. In addition, the authors could have also explained why they didn't collect additional data since the time of the first review. At a minimum, it would be helpful to see 95% confidence intervals plotted around each effect so that the reader has a better impression about the uncertainty associated with the effect (e.g., Ingre, 2013, Neuroimage; Lindquist et al., 2013, Neuroimage).

In order to increase the generality and reliability of our neuroimaging results, we collected fMRI data from a new cohort of female participants, thus addressing both concerns about

sample size and about gender differences. We scanned 15 female participants, with one excluded due to data corruption. This increased our sample size to 32 participants.

We expected this addition to indicate which results were robust and generalised to a bigger and more varied sample size. We found that most results held, others were refined, and others did not generalize and were consequently removed from the manuscript.

We first examined the effect of ‘Selection by Client’ on rTPJ activity. We found that this effect increased in its significance (Figure 4, page 12). We examined the effect size for males only and females only, and the combined effect, and found that while this effect was not as strong in the females’ cohort, it was still evident and enhanced the overall effect (males only Effect Size = 0.79, Beta values (Mean \pm SEM) 6.47 ± 1.40 ; females only: Effect Size = 0.58, Beta values 4.3 ± 2.4 ; combined: Effect Size = 0.68, Beta values 5.52 ± 1.31 $p=0.0005$). In addition, correlation between FNE scores and Selection by Client effect in the rTPJ followed a similar pattern, with overall increase in significance (Correlation: Males only $R^2 = 0.52$, females only $R^2 = 0.14$, combined $R^2 = 0.39$, $p = 0.0004$).

We also plotted the beta values from this region, and indicated the 95% confidence intervals instead of the SEM. It should be noted that, as this is a within-subjects design, the confidence intervals for each condition cannot be directly used to examine differences between the conditions, as comparisons are always paired. Confidence intervals can therefore be useful for demonstration only.

We then examined the effect of trial-by-trial prognostic value comparison in the MPFC. We observed a similar response in the larger cohort (see GLM1 in the figure below). However, we added the trial-by-trial Relative Merit prediction error (PE) as a predictor as well in a new GLM (GLM2), and found that it better explained the activity in the MPFC. The Relative Merit PE is the difference in each trial between the current prognostic value comparison and the previous Relative Merit:

$$RelativeMeritPE(t) = \Delta PrognosticValue(t) - RelativeMerit(t-1)$$

And we now can rewrite the update rule:

$$RelativeMerit(t+1) = RelativeMerit(t) + \gamma \cdot RelativeMeritPE(t)$$

The effect size for Relative Merit PE in the mPFC for males only group was 0.96 (Beta Values (Mean \pm SEM) 0.60 ± 0.15), for females only group the effect size was 0.59 (Beta = 0.32 ± 0.15). The combined group effect size was 0.78 (Beta = 0.48 ± 0.12 , $p = 0.0001$) (Figure 5, page 13).

GLM 1:
Prognostic Value Comparison Only

Trial-by-Trial
Prognostic Value Comparison

GLM 2:

Prognostic Value Comparison + Relative Merit Prediction Error

Trial-by-Trial
Prognostic Value Comparison

Relative Merit
Prediction Error

As relative merit prediction error better explained the responses in MPFC, we used prediction error predictors when examining activity in the Ventral Striatum (VS). We had previously observed BOLD responses in the VS to Relative Merit (positive>negative) and to Selection by Client (chosen>ignored) in a sequential temporal order. We now added the Relative Merit PE and 'Client Selection Switch' predictor as regressors. The 'Client Selection Switch' predictor was set to +1 on trials when the client switched from the other adviser to the participant, to -1 when switching from the participant to the other adviser, and to zero when client selection was the same as in the previous trial.

We found that these prediction error regressors ('Client Selection Switch' and 'Relative Merit PE') did a better job of explaining the variance in VS activity than the Relative Merit or the Selection by Client when included in the same GLM. The temporal order of responses was also consistent with our previous report, i.e. first responding to changes in relative merit and then to changes in client preference. This effect was apparent in both left and right VS., but only the right VS activation survived cluster size correction (compare Figures 4, 5 and S8).

The effect size for Relative Merit PE in VS was 0.89 for males only group (Beta Values (Mean \pm SEM) 0.36 ± 0.09), 0.47 for the females only (Beta Values 0.35 ± 0.20) and 0.62 for the combined group (Beta Values 0.35 ± 0.10 , $p = 0.0013$). The effect size for Client Switch was 0.88 for males (Beta Values 1.81 ± 0.47), 0.35 for females (Beta Values 0.42 ± 0.35) and 0.65 for the combined data (Beta Values 1.21 ± 0.33 , $p = 0.0008$).

As can be seen, increasing the sample size by addition of the female participants increased the findings robustness and reliability.

The additional new participants did not support the PPI analysis results (Figure 6 in the previous version of the manuscript), and the effect of outcome PE in the VS (previously in Supplementary figure S7). We discuss the results of the PPI analysis in details in our answer to the reviewer's comment below, as the revised PPI analysis suggested by the reviewer weakened the strength of the effect in the original cohort, and rendered the effect insignificant when both cohorts were analysed.

In addition, the effect of outcome PE (vicarious reward) in VS, previously reported in Supplementary Materials, was attenuated with the addition of new samples. This finding did not play an important role in our results even before the addition of the neuroimaging participants. In addition, our task was not designed to disentangle vicarious rewards from simple accuracy tracking, as noted by the reviewer in the previous round of revisions. We therefore decided to remove this finding from this work, and plan to examine these effects directly in a dedicated study.

Increasing the sample size has increased our confidence in the reported findings, and hopefully it will help to increase the impact of this work.

Please avoid any points about lateralization unless it is explicitly tested. It doesn't seem to be an important issue at all for this study, so I suggest completely avoiding it.

In the new analysis we did not observe any laterality effect. While only the right VS clusters survived cluster size correction in both Relative Merit PE and Client Selection Switch effects, the unthresholded maps revealed a similar pattern of responses in left VS. We have now followed the reviewer's comment and have avoided the lateralization issue. We present the thresholded maps in the manuscript (Figure 5), but also link to the unthresholded map in NeuroVault, and in the supplementary figures (Figure S6).

The addition of the PPI analysis helps illustrate how these regions are interacting in various parts of the task. However, the description of the PPI model wasn't completely clear to me. It seems like the authors only had one PPI regressor, which wouldn't be adequate to capture the conditions in this design. Please consider showing that the results hold with the generalized PPI model (McLaren et al., 2012, Neuroimage). Also note that similar interactions between MPFC and TPJ were reported in a recent meta-analysis of PPI studies (Smith et al., 2016, HBM), which might lend greater confidence to the authors observed effects.

We ran the PPI analysis again according to the method described in McLaren et al. 2012 (McLaren, Ries, Xu, & Johnson, 2012). This more rigorous analysis revealed a weaker effect in our original male only cohort (N=18), and we observed no effect in the combined dataset (N=32). We removed this observation from the manuscript and focus only on effects that were robust and generalised across genders. We thank the reviewer for directing us to the more rigorous generalized PPI method.

I'm glad the authors clarified the procedures and rationale underlying the noise periods in the task. However, it is not clear to me how "omitting the noise" from the analyses addresses my original point about the noise driving behavior. Please explain further.

In the original comment the reviewer was worried about the noise procedure driving behaviour, stating that, according to Figure 2C, noise periods signalled when the advisers should adjust their confidence. The reviewer then wondered if the pattern of behaviour would persist if the noise periods were omitted. Here we try to clarify our position, and explain why it is unlikely that the entire pattern of behaviour is driven by the schedule of the noise periods.

During the noise periods the evidence displayed to the affected adviser was weaker (ratio of black to white squares in the grid were closer to 50:50) than for the unaffected adviser. This procedure was unknown to the participant and involved a relatively small number of trials. As a result the affected adviser's confidence and accuracy level tended to be lower than the unaffected adviser's confidence. However, our analysis was not carried out on the advisers' confidence directly, but on the advice deviance, i.e. the deviance of confidence from the observed evidence. As long as an observer reduces her confidence adequately to reflect the higher uncertainty in noisy information, her advice deviance would not be expected to change. The behavioural findings therefore still hold, as we showed by comparing the analysis with and without noise periods.

Another important point is that, even though the averaged pattern of behaviour displayed in Figure 2C, shows that changes in advice behaviour are correlated with administration of noise, the individual fluctuations were much more variable. Noise periods were only one source of trial-by-trial fluctuations in confidence and accuracy. For each participant, the order

of evidence observation across trials was randomised. In addition, the coin location on each trial was randomly generated according to the evidence (evidence only implied the probability of the coin location). This meant that each participant experienced individual periods of high/low accuracy and selection by client, which were not time locked (unlike our noise manipulation). These were 'averaged out' of the figure, but strongly influenced behaviour, as was revealed by the individual trial-by-trial model based analysis (see added clarification in the Methods section, page 24).

Reviewer #3 (Remarks to the Author):

I applaud that the authors ran a subsequent study that includes all real participants, as I think this strengthens the manuscript significantly, however I still have a number of concerns.

First, I could not find many of the revisions in the manuscript that the authors said they did. For example, the authors say that they substantially changed the introduction to better address the theoretical motivations for the mpfc and tpj. This was not done. The *only* addition is on page 2 where they write "...evaluation of other's performance and comparing it with that of oneself, a computation underpinned by the mPFC." But *why* cite the mPFC here? Simply referencing this brain region does not mean that it has been properly motivated. In line with this, the authors also state that they now integrated the previous work I referenced in my first round of reviews as relevant to their topic. Again, I cannot find where this was done.

We have revised the introduction again, in order to better motivate our predictions for the cognitive and neural mechanism of strategic advice giving. We have highlighted the revised parts of the manuscript to enhance readability (see highlight section of Introduction, page 4). As the question of advice giving, in contrast to advice taking, has been less studied in the cognitive neuroscience literature, we had to cover in the introduction the hypothesised theoretical predictions for behaviour, in addition to the motivation for neural mechanisms underlying behaviour. In the interest of brevity, we covered only some of the literature on the cognitive mechanisms underlying strategic social behaviour in the introduction, and revisited the subject in the discussion section (highlight section, pages 15-16).

In the previous round of review the reviewer suggested adding literature on advice taking and giving, suggesting the work by Behrens et al. 2008 and Mobbs et al. 2015. These works and others are cited in the introduction and are now highlighted in the manuscript (green highlights, page 4).

Second, it is still not clear to me why the authors are arguing that in a modelling paper they do not want to fit their learning rates to the MRI data?

In the model based fMRI approach employed here (Daw, 2011; O'Doherty, Annals 2017), we used a computational model to infer unobservable time-varying variables, such as the fluctuations in Relative Merit, by fitting our model to the behavioural data. The model includes a number of free parameters, which are fixed for each participant and do not vary over time. One such parameter is the learning rate, which governs how fast the model learns from experience (i.e. from each prognostic value comparison between advisers). The learning rate (and the other parameters) is assumed to be stable on trial-by-trial basis (although different across subjects). Thus, we cannot use the learning rate as a parametric modulator in a first level fMRI analysis because it does not fluctuate on a trial-by-trial basis. It would have been possible with different computational models in different experimental designs, where the learning rate fluctuates on a trial-by-trial basis (see McGuire et al. 2014 for an example of how learning rates could be neurally instantiated).

On the other hand, the model therefore serves to estimate the hidden variable Relative Merit (RM), which does fluctuate on a trial-by-trial basis. The resulting time-series is then taken as a prediction of the BOLD activity at the brain area hypothesised to carry out the computation of Relative Merit. This hypothesis is then tested in the form of a General Linear Model. Thus, by regressing the hidden variables (derived from computational interpretation of the

behaviour) onto the BOLD signal we test if the hypothesised brain area (mPFC) represent the variable assumed by the model.

Importantly, we first determined which variables are relevant for behaviour using a model comparison procedure, and then proceeded to examine their neural correlates as described above. We have now made this explicit in the main text (see cyan highlights in main text page 7 and 13).

In line with this, their ST1 table does not include alphas and it should.

We assume that the alphas the reviewer refers to are the intercepts, in our case the participant's mean advice deviance. The intercept (alpha), in our model, is represented by the 'bias' parameter, which is included in table ST1. When bias parameter is positive, the average advice deviance is positive, which means that the participant tended to be overconfident. We added a clarification in the text (page 8 and in ST1).

Third, the manuscript is still rife with grammar issues. There is about one on every page. For example, on page 2 alone, there is a sentence that reads "...the theory suggests people may response according to their rank in a group." Please read over your manuscript carefully before submitting as these mistakes are easy to fix and only serve to annoy the readers.

All authors and an external reader have proof read the manuscript, and we hope that there are no more grammar issues.

Fourth, I do think we should be staying away from calling a network of brain regions the 'social brain system' (pg 2). That 'social brain system' is the same system that encodes many non-social phenomena as well. Again, this type of language boils down to proper motivation of understanding why certain brain regions may be involved in processing this type of advice giving.

We revised the manuscript and removed the term 'social brain network', and give a more explicit motivation for assigning specific brain areas to distinct cognitive processes (page 2, 4).

Fifth, how was the PPI done? What was the seed region? Again, more details about these analyses must be reported.

The PPI analysis did not fare well after applying the methodological recommendations provided by reviewer 2, and did not generalise to the new mixed gender cohort of participants. We have therefore removed this analysis from the manuscript.

REVIEWERS' COMMENTS:

Reviewer #2 (Remarks to the Author):

The authors have addressed all of my concerns, and the manuscript is much improved. I appreciate that the authors nearly doubled their sample size in response to the reviewers.

Editorial Note: Reviewer #3 was unable to return a re-review in a timely manner, so we asked another reviewer to comment on the authors' response to Reviewer #3's previous concerns. This reviewer approved the revision.